# Many versus one: the disorder operator and entanglement entropy in fermionic quantum matter

Weilun Jiang,[1,2] Bin-Bin Chen,[3,*] Zi Hong Liu,[4] Junchen Rong,[5]
Fakher F. Assaad,[4] Meng Cheng,[6] Kai Sun,[7] and Zi Yang Meng[3,†]

[1]*Beijing National Laboratory for Condensed Matter Physics and Institute of Physics, Chinese Academy of Sciences, Beijing 100190, China*
[2]*School of Physical Sciences, University of Chinese Academy of Sciences, Beijing 100190, China*
[3]*Department of Physics and HKU-UCAS Joint Institute of Theoretical and Computational Physics,*
*The University of Hong Kong, Pokfulam Road, Hong Kong SAR, China*
[4]*Institut für Theoretische Physik und Astrophysik and Würzburg-Dresden Cluster*
*of Excellence ct.qmat, Universität Würzburg, 97074 Würzburg, Germany*
[5]*Institut des Hautes Études Scientifiques, 91440 Bures-sur-Yvette, France*
[6]*Department of Physics, Yale University, New Haven, CT 06520-8120, USA*
[7]*Department of Physics, University of Michigan, Ann Arbor, Michigan 48109, USA*
(Dated: June 8, 2023)

Motivated by recent development of the concept of the disorder operator and its relation with entanglement entropy in bosonic systems, here we show the disorder operator successfully probes many aspects of quantum entanglement in fermionic many-body systems. From both analytical and numerical computations in free and interacting fermion systems in 1D and 2D, we find the disorder operator and the entanglement entropy exhibit similar universal scaling behavior, as a function of the boundary length of the subsystem, but with subtle yet important differences. In 1D they both follow the $\log L$ scaling behavior with the coefficient determined by the Luttinger parameter for disorder operator, and the conformal central charge for entanglement entropy. In 2D they both show the universal $L \log L$ scaling behavior in free and interacting Fermi liquid states, with the coefficients depending on the geometry of the Fermi surfaces. However at a 2D quantum critical point with non-Fermi-liquid state, extra symmetry information is needed in the design of the disorder operator, so as to reveal the critical fluctuations as does the entanglement entropy. Our results demonstrate the fermion disorder operator can be used to probe quantum many-body entanglement related to global symmetry, and provides new tools to explore the still largely unknown territory of highly entangled fermion quantum matter in 2 or higher dimensions.

## I. INTRODUCTION

In recent years, quantum many-body entanglement has become a subject of intense activity, and one fundamental question is to find ways to probe one big, or many different aspects of many-body entanglement in quantum matter. The most familiar probes include non-local measurements such as the von Neumann and Rényi entanglement entropy (EE), the entanglement spectrum (ES) and bipartite fluctuations, which have been studied in many different types of quantum many-body systems [1–22]. Recently, along this line, it has been proposed that the concept of the *disorder operator* may provide an effective tool for capturing some aspects of the entanglement information, especially the interplay with global symmetry [23–31]. In a variety of boson/spin systems, universal quantities, such as the logarithmic corner corrections in (2+1)D conformal field theories (CFT) [16, 25, 26], non-unitary CFT of deconfined quantum criticality [30], and defect quantum dimensions in gapped phases [29], were successfully extracted from the scaling behavior of the disorder operators. Compared to EE, the computational cost of the disorder operators is often significantly reduced, which allows for access to much larger sizes with reduced finite-size effect.

In this paper, we would like to expand upon these developments and demonstrate that, the disorder operator can be utilized to probe many-body entanglement in fermion systems with computational simplicity and reliable data quality. We will show fermion disorder operators obey similar scaling laws as that of EE, but with important differences in that by construction the disorder operator measures the symmetry charge contained in a region, while the EE is not directly related to any global symmetry. In this way, the fermion disorder operator provides a new viewpoint on the entanglement information, that is to say, probing the universal entanglement by nature but also subjected to the symmetry details by construction – and it is in this duality that the fermion disorder operator manifests itself as a new and promising vehicle with which we could explore the territory of quantum entanglement in 2D or higher dimensional interacting fermion systems. We also note that the disorder operator has close experimental relevance in detecting the entanglement information in quantum materials, and holds the possibility to extend to finite temperature to probe the entanglement in mixed states [32].

We begin the narrative with a brief discussion about the development of computational scheme and the known scaling behavior of EE. In general, the Rényi EE can be measured via the replica trick [4] in the path-integral quantum Monte Carlo (QMC) simulations. More concretely, for the $n$-th order Rényi entropy $S_n$, one creates $n$ copies of the system (i.e. replicas), properly connected along the entanglement cut, and measures the correlation between the replicas [33]. Due to the $n$-fold expansion of the configuration space and the nontrivial connectivity between the replicas during the sampling processes, obtaining numerical data of EE with good quality in QMC simulations has remained a very challenging task. De-

———

[*] bchenhku@hku.hk
[†] zymeng@hku.hk

spite the difficulty, we note there are recent progresses in the nonequilibrium incremental algorithm to obtain the $S_n$ with high efficiencies and precisions [15, 16, 18, 34] and similar advances in measuring the ES [17, 19] in QMC simulations, which effectively solve the difficulties in measuring EE in boson/spin systems in (2+1)D (that can be efficiently simulated by QMC methods).

However, progress in the extraction of entanglement information in interacting fermion systems has largely fallen behind. In the fermion QMC simulation for EE, one could likewise calculate the joint probability [11] by constructing the extended manifold for ensemble average [12–14]. But the necessity of using replica renders the already difficult determinant QMC simulations (DQMC, usually the computational complexity scales as $O(\beta N^3)$ with $\beta = 1/T$ and $N = L^d$ for the $d$ spatial dimension even heavier, and such computational burden has hinged the usage and discussion of the scaling behavior of EE in the exploration of the highly entangled fermionic quantum matter. We note the recent progresses in this regard, with better data quality and the approximate $O(\beta N^3)$ complexity [20–22].

Likewise in the experimental aspect, it is also a difficult task to probe the EE even for (over-)simplified systems, e.g., quantum point contact model, which is easy to implemented by two conjoint charge reservoirs [10, 35, 36]. The entanglement of such systems comes from the transmitted charges between two reservoirs, and can be measured via the statistics or distribution of charge. To be more specific, the EE is expanded by even cumulants of charges. It is easy to find its correspondence with physical observables – i.e. the disorder operators discussed here. However, obstacles remain when in face of more complicated experimental device, where the single degree of freedom is hard to describe the whole entanglement. Therefore, a proper witness of entanglement remains debatable. In any case, the disorder operator serves as a useful tool that may guided the experimental measurements. Such bipartite fluctuations [10, 36] have also been studied numerically in (1+1)D conformal field theory, 1D Luttinger liquid, and Fermi liquid systems. Nevertheless, for more complicated systems, e.g. 2D quantum critical point (QCP) and non-Fermi liquid, systematic investigations of bipartite fluctuation are still missing.

We also notice the recent development on the symmetry resolved entanglement. Specifically, in the quantum systems with a globally conserved charge, the entanglement can be decomposed into different symmetry sectors, revealing the microscopic structure and phase transitions of the system. The pioneering work by Goldstein *et al.* [37] presents a geometric approach for extracting the contribution from individual charge sectors to the subsystem's entanglement basing on the replica trick method, via threading appropriate conjugate Aharonov-Bohm fluxes through a multisheet Riemann surface. Subsequent studies focused on various applications on certain systems, including fermionic system [38, 39], spin system [40], quantum field theory [41, 42], CFT [43] and topological matters [44]. All of them provide a new perspective for the study of entanglement entropy. But these results are still mainly in 1D or free system, different from the 2D interacting

fermionic systems we are focused here with disorder operator.

In this work, we show that the disorder operator offers a numerically easier way to access certain aspects of quantum many-body entanglement in interacting fermionic systems. We design the disorder operators to probe the charge and spin fluctuations in the entangled subregion and show that they exhibit similar scaling behavior as that of EE. In fact, we show for non-interacting fermions (Rényi) EE can be exactly expressed in terms of charge disorder operators. However, in contrast to the EE measurement with additional numerical complexities due to replicas, the disorder operator is an equal-time observable with no need for extended manifold and substantially reduces the computational complexity. We find the fermion disorder operators share the good properties of its bosonic cousins [25, 26, 29, 30] and demonstrate their applications in several representative classes of fermionic systems, including the free Fermi surface (FS) systems, Luttinger liquid (LL) in 1D, and a 2D lattice model of spin-1/2 fermions interacting with Ising spins which exhibits both the Fermi liquid (FL) phases and non-Fermi liquid (nFL) QCP separating symmetric and symmetry-breaking FLs.

## II. SUMMARY OF KEY RESULTS

We start with a list of the main innovation points of this work.

1. We give the exact relation between Rényi EE and the disorder operator in the non-interating fermionic systems.

2. We find the disorder operator serves as a more powerful tools to extract Luttinger parameter compared with traditional methods in 1D interacting fermionic systems.

3. We investigate the 2D nFL and FL by means of the two types of disorder operators, and uncover their connection with the entanglement entropy.

Below, we explain these points in a more pedagogical perspective that resonate with the remaing sections of the paper.

For free fermions, we show that the charge/spin disorder operator directly measures the quantum entanglement. For example, the second-order Rényi entropy $S_2$ is identical to the logarithm of the charge (spin) disorder operator (defined below) at angle $\theta = \frac{\pi}{2}$ ($\theta = \pi$):

$$S_2 = -2\log\left|X^\rho\left(\frac{\pi}{2}\right)\right| = -2\log|X^\sigma(\pi)|. \tag{1}$$

More general relations between Rényi entropies $S_n$ and the disorder operators are given in Eqs. (8)- (10) in Sec. IV A. For a non-interacting Fermi liquid ground state, it is well-known that $S_2 \sim L_M^{d-1}\log L_M$, where $d$ is the spatial dimension and $L_M$ is the linear size of the subregion $M$ used to define the disorder operator and/or the EE. For generic values of $\theta$, the charge/spin disorder operator obeys the same functional form. The coefficient of the $L_M^{d-1}\log L_M$ term, which is proportional to $\theta^2$, is fully consistent with analytic form based on the Widom-Sobolev formula [Eq.(15)] [45–48].

For interacting fermions in 1D, we find that both Rényi entropies $S_n$ and the logarithm of the disorder operators $-\log|X^\rho(\theta)|$ follow the same form of $\log L_M$, same as the free fermion system, but with different coefficients. It is well-known that the coefficient of the $\log L_M$ term in EE measures the conformal central charge. We will find that the coefficient of the charge/spin disorder operator gives the Luttinger parameter in the charge/spin sector. Utilizing density-matrix renormalization group (DMRG) and DQMC simulations, we discover that disorder operator offers a highly efficient way to measure the Luttinger parameter. In comparison to more conventional approaches based on structure factors at small momentum, the disorder operator exhibits much less finite-size effect, and can provide highly accurate estimates of Luttinger parameters with moderate numerical costs.

For 2D interacting fermions, we study an itinerant fermion systems, whose phase diagram shows a continuous quantum phase transition between a paramagnetic phase and an Ising ferromagnetic phase [49–52]. Away from the QCP, the system is a FL, while in the quantum critical regime, critical fluctuations drive the system into a nFL [51, 53–55]. We compute the charge and spin disorder operators, as well as the second Rényi entropy using DQMC. The simulations indicate that within numerical errorbars, $S_2$ and $-\log|X_M^\rho|$ obey the same form of $\sim L_M \log L_M$ as non-interacting fermions. For the spin disorder operator, as the total magnetization $S^z$ is the order parameter of this quantum phase transition, the coefficient of the $L_M \log L_M$ term appears to diverge near the QCP, as expected due to the divergent critical fluctuations. As for the charge disorder operator $-\log|X_M^\rho|$ and $S_2$, no singular behavior is observed near the QCP. Instead, the values of $-\log|X_M^\rho|$ and $S_2$ remain very close to the free-fermion formula [Eq.(15)], and the deviation is less than a couple percent.

Although the observed deviation from the non-interacting case is small, it is not zero. More careful analysis reveals that as interactions becomes stronger (getting close to the QCP), the interaction effect increases the value of $S_2$ and decreases the value of $-\log|X_M^\rho|$. This effect is beyond the numerical error bar, and more importantly, finite-size scaling analysis shows that this deviation does not disappear as the system size increases, suggesting that it may survive in the thermodynamic limit. More precisely, deviations of $-\log|X_M^\rho|$ from the results of the free system appear to saturate at large system size, both in the FL phase and in the nFL near the QCP. This observation indicates that $-\log|X_M^\rho|$ still scales as $L_M \log L_M$, same as the non-interacting Fermi sea, but the coefficient of the $L_M \log L_M$ term decreases as interactions becomes stronger, both in the FL phase and at the QCP. For $S_2$, away from the QCP, its deviation from the free theory saturates at large system size, and thus we expect the relation $S_2 \sim L_M \log L_M$ survives in the FL phase, with a coefficient increasing as interaction gets stronger. At the QCP (i.e., a nFL), the deviation of $S_2$ from the non-interacting value does not seem to saturate, up to the largest system size that we can assess. Due to the limitation of system sizes, we cannot pinpoint the precise form of $S_2$ at the QCP. If the deviation eventually saturates as the system size increases further, it would

imply that $S_2 \sim L_M \log L_M$ at the QCP, although with a significant enhancement of the coefficient. If the deviation keeps increasing with larger system sizes, it means $S_2$ grows faster than $L_M \log L_M$ at large $L_M$. Our findings offer plentiful opportunities for future investigations of the entanglement information of interacting fermion systems at 2D and higher dimensions.

### III. THE DISORDER OPERATOR

In a fermionic system with conserved particle number, the corresponding charge U(1) symmetry allows us to define a charge disorder operator

$$\hat{X}_M^\rho(\theta) = \prod_{i \in M} e^{i\theta \hat{n}_i} = e^{i\theta \hat{N}_M}, \tag{2}$$

where $\hat{n}_i$ is the particle number operator at lattice site $i$, and $\hat{N}_M$ is the total particle number operator of region $M$, as shown in the schematic plots in Fig. 3 (b) and (d).

If the system is composed of spin-1/2 fermions and has an U(1) spin conservation, e.g., spin rotations around the $z$ axis, we can further define a spin disorder operator

$$\hat{X}_M^\sigma(\theta) = \prod_{i \in M} e^{i\theta \hat{S}_i^z} = e^{i\theta \hat{S}_M^z}, \tag{3}$$

where $\hat{S}_i^z = \frac{1}{2}(\hat{n}_{i\uparrow} - \hat{n}_{i\downarrow})$ is the $z$-component of the spin operator at lattice site $i$, and $\hat{S}_M^z$ is the total spin $z$-component of region $M$.

For convenience, we denote $X_M(\theta) = \langle \hat{X}_M(\theta) \rangle$, where $\langle \cdot \rangle$ is the ground state expectation value. The computation of $X_M$ as an equal-time observable in DQMC is discussed in the Sec. I of the Supplemental Material (SM) [56]. It is straightforward to prove that if the system is divided into two parts $M$ and $\overline{M}$, the ground state preserves the symmetry, $|X_M^{\rho/\sigma}(\theta)| = |X_{\overline{M}}^{\rho/\sigma}(\theta)|$, in analogy with EE. In addition, the disorder operator also obeys the following relations: $X_M^\rho(\theta) = X_M^\rho(\theta + 2\pi)$, $X_M^\sigma(\theta) = X_M^\sigma(\theta + 4\pi)$, $X_M^{\rho;\sigma}(\theta) = X_M^{\rho;\sigma}(-\theta)^*$, and $X_M^{\rho/\sigma}(0) = 1$. In the small $\theta$ limit, $X_M^{\rho/\sigma}$ measures the density/spin fluctuations in the region $M$: $-\log|X_M^\rho(\theta)|/\theta^2 = \langle(\hat{N}_M - \langle\hat{N}_M\rangle)^2\rangle$ [26] and $-\log|X_M^\sigma(\theta)|/\theta^2 = \langle(\hat{S}_M^z - \langle\hat{S}_M^z\rangle)^2\rangle$.

For non-interacting fermions, the disorder operators can be calculated using the equal-time Green's function

$$G_{i\sigma,j\sigma'} = \langle \hat{c}_{i\sigma} \hat{c}_{j\sigma'}^\dagger \rangle, \tag{4}$$

where $\hat{c}_{i\sigma}$ and $\hat{c}_{j\sigma'}^\dagger$ are fermion annihilation and creation operator at sites $i$ and $j$ respectively, and $\sigma$ and $\sigma'$ are the spin indices. For the sake of simplicity, we define the matrix $G_M = G_{i\sigma,j\sigma'}$ with sites $i$ and $j$ being confined in subregion $M$. The charge disorder operators for the region $M$ can be written as a matrix determinant

$$X_M^\rho(\theta) = \det\left[G_M + (\mathbb{1} - G_M)e^{i\theta}\right], \tag{5}$$

where $\mathbb{1}$ represents the identity matrix. For spin-1/2 fermions with U(1) spin conservation, the Green's function matrix is block-diagonal with two sectors, i.e., the spin up sector $G_{M,\uparrow}$ and the spin down sector $G_{M,\downarrow}$, and thus the disorder operator can be written as

$$X_M^\rho(\theta) = X_{M,\uparrow}(\theta) X_{M,\downarrow}(\theta) \tag{6}$$

$$X_M^\sigma(\theta) = X_{M,\uparrow}\left(\frac{\theta}{2}\right) X_{M,\downarrow}\left(-\frac{\theta}{2}\right) \tag{7}$$

where $X_M^\sigma(\theta) = \det\left[G_{M,\sigma} + (\mathbb{1} - G_{M,\sigma})e^{i\theta}\right]$ for spin index $\sigma = \uparrow, \downarrow$. This gives the simple relation between spin and charge disorder operators $|X_M^\rho(\theta)| = |X_M^\sigma(2\theta)|$ in the non-interacting systems, as we now turn to.

## IV.  NON-INTERACTING LIMIT

In this section, we study non-interacting systems, where an explicit connection between the disorder operator $X_M^\rho(\theta)$ and the Rényi entropy $S_n$ can be analytically proved. Besides the mathematical proof, we will also demonstrate and verify these relations numerically in concrete models, before exploring interacting systems.

### A.  Exact relations between disorder operator and EE

For non-interacting systems, the Rényi entropy $S_n$ can be exactly related to the disorder operator $X_M^\rho(\theta)$. For $S_2$ and $S_3$, we have

$$S_2 = -2\log\left|X_M^\rho\left(\frac{\pi}{2}\right)\right|, \tag{8}$$

$$S_3 = -\log\left|X_M^\rho\left(\frac{2\pi}{3}\right)\right|. \tag{9}$$

For general values of $n$,

$$S_n = \frac{1}{1-n}\sum_{\alpha=0}^{n-1}\log X_M^\rho\left(\left[\frac{1+(-1)^n}{2}+2\alpha\right]\frac{\pi}{n}\right)$$
$$= \frac{2}{1-n}\sum_{\alpha=0}^{2\alpha+1<n}\log\left|X_M^\rho\left(\left[\frac{1+(-1)^n}{2}+2\alpha\right]\frac{\pi}{n}\right)\right|. \tag{10}$$

In the presence of U(1) spin conservation, these relations also apply to spin disorder operators $X_M^\sigma$, but the values of $\theta$ change, e.g., $S_2 = -2\log|X_M^\sigma(\pi)|$.

Detailed proof of these relations can be found in Sec. II of the SM [56]. Here, we just briefly outline the idea. It has been shown that the Rényi entropy can be calculated using the cumulants of charge fluctuations $C^k = (-i\partial_\theta)^k \log X_M^\rho(\theta)|_{\theta=0}$ [10, 57], which is the key reason why disorder operators can give the value of the Rényi entropy. In addition, to obtain a simple relation between $S_n$ and $X_M^\rho$, we express both these two quantities in terms of equal-time fermion Green's functions $G_{i\sigma,j\sigma'}$, utilizing Wick's theorem for $S_n$ [10, 11] and the Levitov-Lesovik determinant formula [36, 58, 59] for $X_M^\rho$. The detailed derivation is given in

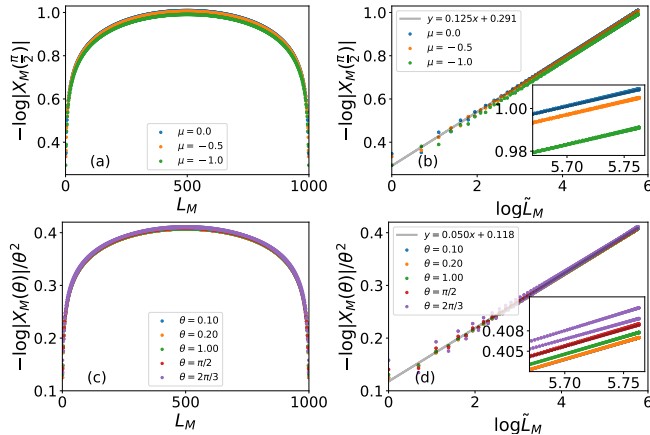

FIG. 1.  (a) and (b) The disorder operator $-\log|X_M(\frac{\pi}{2})|$ of a 1D free fermion chain at various $\mu$. Here we use the PBC and the system size is $L = 1000$. In (b), we fit the data with $-\log|X_M(\frac{\pi}{2})| = s_{1D}(\frac{\pi}{2})\log\tilde{L}_M + \text{const}$, where $\tilde{L}_M = \frac{L}{\pi}\sin\frac{\pi L_M}{L}$ is the conformal distance. The grey line shows the best fitting for $\mu = 0$ at $\log\tilde{L}_M > 2$. Upon varying $\mu$, we find that the constant part in the fitting varies [see the inset panel in Fig.(b)], but the value of $s_{1D}(\frac{\pi}{2})$ is universal $\sim 0.125$, in full agreement with theory prediction $s_{1D}(\frac{\pi}{2}) = 1/8$. In (c) and (d), we plot $-\log|X_M(\theta)|/\theta^2$ at various $\theta$ for $\mu = 0$. In the large size limit, the leading order contribution $\sim\log\tilde{L}_M$ is found to be independent of $\theta$. In panel (d), the grey line shows the best fitting for $\theta = 0.1$ at $\log\tilde{L}_M > 2$, which gives the slope 0.050, in good agreement with the theory prediction $\frac{1}{2\pi^2} \approx 0.051$.

SM [56]. Besides, the relation between the reduced density matrix with the equal time Green's functions is deduced in Ref. [60]. Refer to this equation, $S_n$ is expressed with Green's functions in Ref. [11].

### B.  The disorder operator and EE for 1D non-interacting fermions

In this subsection, we demonstrate the scaling behavior of the disorder operators in a 1D non-interacting fermion model with periodic boundary conditions (PBCs). For simplicity, we focus on spinless fermions, and utilize Eq. (5) to compute the disorder operator. This calculation can be easily generalized to non-interacting spin-1/2 fermions with U(1) spin conservation, where the charge and spin disorder operator $X_M^\rho(\theta) = X_M^\uparrow(\theta)X_M^\downarrow(\theta)$ and $X_M^\sigma(\theta) = X_M^\uparrow(\frac{\theta}{2})X_M^\downarrow(-\frac{\theta}{2})$, and each spin sector can be treated as spinless fermions.

The Hamiltonian of this model is

$$\hat{H} = -t_1\sum_{\langle i,j\rangle}\hat{c}_i^\dagger\hat{c}_j - \mu\sum_i\hat{n}_i, \tag{11}$$

where the nearest-neighbor hopping strength $t_1$ is set to unity and the chemical potential $\mu$ is between $\pm 2$ to ensure a partially filled band. We choose $M$ to be an interval of the 1D chain with length $L_M$, and plot $-\log|X_M(\theta)|$ as a function of $L_M$ in Fig. 1. Because $|X_M| = |X_{\overline{M}}|$, the function is necessarily symmetric with respect to $L_M = \frac{L}{2}$, with

$L$ being the system size [Fig. 1(a) and (c)]. In the regime $a \ll L_M \ll L$, where $a$ is the lattice constant, the leading term of $-\log|X_M|$ exhibits a universal scaling relation $-\log|X_M| = s_{1D}(\theta)\log L_M + O(1)$, and the data suggests that the coefficient $s_{1D}(\theta)$ is independent of the chemical potential $-2 < \mu < 2$.

In order to obtain a more accurate fitting for the coefficient $s(\theta)$, here we introduce the conformal distance $\tilde{L}_M = \frac{L}{\pi}\sin\frac{\pi L_M}{L}$, and replace the fitting form $-\log|X_M| = s_{1D}(\theta)\log L_M + O(1)$ with $-\log|X_M| = s_{1D}(\theta)\log\tilde{L}_M + O(1)$. In the regime of $a \ll L_M \ll L$, $\tilde{L}_M$ coincides with $L_M$ and thus the two fitting formulas are interchangeable. However, as $L_M$ approaches $\frac{L}{2}$ or becomes larger than $\frac{L}{2}$, the introduction of the conformal length greatly suppresses non-universal subleading terms and thus provides a much more accurate value for $s_{1D}$. As shown in Fig. 1(b) and (d), the relation $-\log|X_M| \sim s_{1D}(\theta)\log\tilde{L}_M$ holds for a very wide region from $L_M$, except for the data points with $L_M$ very close to 1 or $L_M$. For the coefficient $s_{1D}$, in principle, it can be written as a power-law expansion $s_{1D}(\theta) = S_2\theta^2 + S_4\theta^4 + \ldots$, where the coefficients $S_k$ can be obtained from the corresponding cumulants of charge fluctuations $C^k = (-i\partial_\theta)^k \log X_M^\rho(\theta)|_{\theta=0}$ [10, 57]. The coefficient $S_2 = \frac{1}{2\pi^2}$ can be evaluated exactly via density-density correlations. As for higher order terms ($S_k$ with $k > 2$), the Widom-Sobolev formula suggests that they shall all vanish, $S_k = 0$ for $k > 2$ [57]. Therefore, we shall expect

$$-\log|X_M(\theta)| = \frac{\theta^2}{2\pi^2}\log L_M + O(1) \qquad (12)$$

for $-\pi < \theta < \pi$. For $\theta > \pi$ or $\theta < -\pi$, the value of $-\log|X_M(\theta)|$ can be obtained using the periodic condition $X_M(\theta) = X_M(\theta + 2\pi)$.

As shown in Fig. 1(b) and (d), the numerical fitting indeed supports the analytic result and the Widom-Sobolev formula. We find that $s_{1D}(\theta)/\theta^2$ is a constant $\sim 0.050$, independent of the value of $\theta$, and this value is very close to the theoretical expectation $S_2 = \frac{1}{2\pi^2} \approx 0.051$.

To further verify Eq. (12), we use this formula to compute the Rényi EE. By plugging in Eq. (12) into Eqs. (8)-(10), we find that $S_2 = \frac{1}{4}\log L_M + O(1)$, $S_3 = \frac{2}{9}\log L_M + O(1)$, and $S_n = \frac{1}{6}(1 + \frac{1}{n})\log L_M + O(1)$. This result fully agrees with the Rényi entropy formula of 1D free fermions $S_n = \frac{c}{6}(1 + \frac{1}{n})\log L_M + O(1)$, where $c = 1$ is the conformal central charge of 1D spinless fermions.

### C. Disorder operator and EE for 2D non-interacting fermions

In this section, we study 2D free fermions with FS. Here we use a square lattice model with nearest-neighbor ($t_1$) and next-nearest-neighbor hoppings ($t_2$). The Hamiltonian is

$$\hat{H} = -t_1\sum_{\langle i,j\rangle}\hat{c}_i^\dagger\hat{c}_j - t_2\sum_{\langle\langle i,j\rangle\rangle}\hat{c}_i^\dagger\hat{c}_j - \mu\sum_i\hat{n}_i \qquad (13)$$

and we choose a square subregion $M$ as shown in Fig. 3(d).

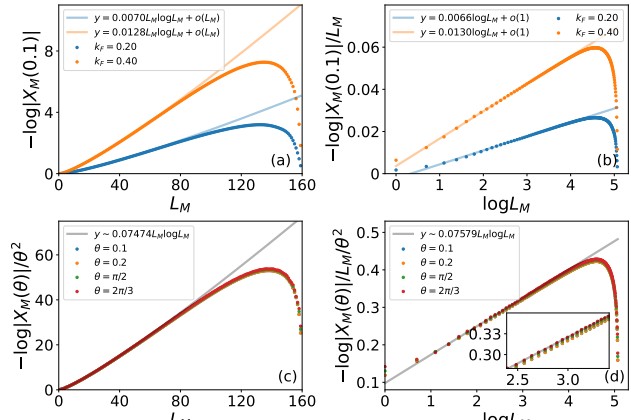

FIG. 2. (a) and (b) The disorder operator $-\log|X_M(\theta = 0.1)|$ of a 2D non-interacting Fermi gas with Fermi wavevector $k_F = 0.2$ (or $k_F = 0.4$). The model utilizes a $L \times L = 160 \times 160$ square lattice with PBCs. Here, we fit the data with $y = s_{2D}L_M\log L_M + bL_M + c$ in (a) and $y = s_{2D}\log L_M + b$ in (b), which is expected for the regime $1 \ll L_M \ll L$. Here, we used points with $L_M \in [10, 30]$ for the fitting. The value of $s_{2D}$ obtained from the fitting is consistent with the analytic analysis $s_{2D} = k_F\theta^2/\pi^3$. (c) $-\log|X_M(\theta)|/\theta^2$ with $t_1 = 1, t_2 = 0, \mu = -0.5$ for various $\theta$s. The two sets of data overlays on top of each other, indicating $s_{2D} \propto \theta^2$. The solid lines are fittings using $y = s_{2D}L_M\log L_M + bL_M + c$ for (c) and $y = s_{2D}\log L_M + b$ for (d), for points with $L_M \in [10, 30]$ to avoid the finite size effect.

For a 2D FL with a circular-shaped FS, the disorder operator at the small $\theta$ limit can be calculated exactly (See Sec. IV of SM [56])

$$-\log|X_M(\theta)| = \frac{k_F\theta^2}{\pi^3}L_M\log L_M + O(L_M) \qquad (14)$$

where $k_F$ is the Fermi wave-vector. In Fig. 2 (a) and (b), we compare this analytic prediction with numerically obtained $|X_M(\theta)|$. We study the two different cases with a nearly circular FS, one with $k_F \approx 0.4$ ($t_1 = 1, t_2 = 0, \mu = -3.84$) and the other with $k_F \approx 0.2$ ($t_1 = 1, t_2 = 0, \mu = -3.96$), and obtain the coefficients $s(\theta)$ by fitting the data with the formula $-\log|X_M(\theta)| = s_{2D}(\theta)L_M\log L_M + b\log L_M + c$. Although the FS deviates slightly from a perfect circle, the fitted values of $s_{2D}(\theta)$ are very close to the analytic formula Eq (14).

In general, when the FS is not a perfect circle but still share the same topology, the disorder operator can be calculated using the Widom-Sobolev formula [45–48, 57],

$$-\log|X_M(\theta)| = \frac{\theta^2}{8\pi^2}\lambda_M L_M^{d-1}\log L_M + O(L_M^{d-1}) \qquad (15)$$

where the prefactor $\lambda_M$ is defined as

$$\lambda_M = \iint \frac{dA_x dA_k |n_x \cdot n_k|}{(2\pi)^{d-1}}. \qquad (16)$$

Here $n_x$ and $n_k$ are unit normal vectors of the real space boundary of $M$ and the momentum space boundary of FS,

respectively. The double integral goes over the boundary of $M$ and the boundary of the Fermi sea. For a circular FS, this formula recovers Eq. (14) above. Note that $\lambda_M$ is a pure geometric quantity, determined by the shape of $M$ and the FS.

Our numerical fitting is in full support the Widom-Sobolev formula. In the regime $a \ll L_M \ll L$, we find $-\log|X_M(\theta)| \sim s_{2D}(\theta) L_M \log L_M$ and $s_{2D}(\theta) \propto \theta^2$ [Fig. 2 (c) and (d)], and the constant ratio $s_{2D}(\theta)/\theta^2$ is in agreement with the integral in Eq. (15). In addition, we have also verified the connection between $|X_M(\theta)|$ and the Rényi EE. Using Eq. (15) and Eqs. (8)- (10), we get

$$S_2 = \frac{\lambda_M}{16} L_M \log L_M + O(L_M), \qquad (17)$$

$$S_3 = \frac{\lambda_M}{18} L_M \log L_M + O(L_M), \qquad (18)$$

$$S_n = \frac{1 + n^{-1}}{24} \lambda_M L_M \log L_M + O(L_M). \qquad (19)$$

This result is fully consistent with free-fermion EE obtained from the Widom-Sobolev formula [45–48, 57, 61, 62].

## V. DISORDER OPERATOR IN INTERACTING FERMION SYSTEMS

For interacting fermions, the exact relation between EE and disorder operator [Eqs. (8)- (10)] is no longer valid, but there still exists interesting connection between these two quantities in their implementation in DQMC simulations.

### A. DQMC relation between disorder operator and EE

To demonstrate this connection, here we use an auxiliary field $\{s\}$ to decouple the interactions between fermions such that we transform the interacting fermion problem into an equivalent model where fermions couple to this auxiliary field $\{s\}$, which mediate interactions between fermions. This is how DQMC is implemented to simulate interacting fermion models. In this setup, the expectation value of a physics quantity can be written as $\langle O \rangle = \sum_{\{s\}} P_s \langle O \rangle_s$, where $\sum_{\{s\}}$ sums over all auxiliary field configurations; $P_s$ is the probability distribution of auxiliary field configurations; and $\langle O \rangle_s$ can be viewed as the expectation value of $O$ for an static auxiliary field configuration $s$.

Here, we focus on the relation between $S_2$ and $X_M^\rho(\frac{\pi}{2})$ [Eq. (8)]. Using the auxiliary field approach shown above, the disorder operator can be written as

$$X_M^\rho(\theta) = \sum_{\{s\}} P_s \det\left[ G_{M,s} + (\mathbb{1} - G_{M,s})e^{i\theta} \right], \qquad (20)$$

where $G_{M,s}$ is the equal-time fermion Green's function for the auxiliary field configuration $s$. In the non-interacting limit, there is no need to introduce the auxiliary field, i.e., $G_M$ is independent of $s$ and thus Eq. (20) recovers the free fermion formula Eq. (5) at $\theta = \frac{\pi}{2}$.

Utilizing Eq. (20), we can write down the following formula for interacting fermions

$$-2\log\left|X_M^\rho\left(\frac{\pi}{2}\right)\right| = -\log\{\sum_{\{s\}} P_s \det\left[G_{M,s} + i(\mathbb{1} - G_{M,s})\right] \times \sum_{\{s'\}} P_{s'} \det\left[G_{M,s'} - i(\mathbb{1} - G_{M,s'})\right]\}$$

$$= -\log\{\sum_{\{s,s'\}} P_s P_{s'} \det\left[G_{M,s} G_{M,s'} + (\mathbb{1} - G_{M,s})(\mathbb{1} - G_{M,s'}) + i(G_{M,s} - G_{M,s'})\right]\} \qquad (21)$$

where $\{s, s'\}$ label independent auxiliary field configurations. For the Rényi entropy $S_2$, as shown in Ref. [11], we have

$$S_2 = -\log\{\sum_{\{s,s'\}} P_s P_{s'} \det\left[G_{M,s} G_{M,s'} + (\mathbb{1} - G_{M,s})(\mathbb{1} - G_{M,s'})\right]\}. \qquad (22)$$

By comparing Eqs. (21) and (22), we can see that $-2\log|X_M^\rho(\frac{\pi}{2})|$ and $S_2$ only differs by one term $i(G_{M,s} - G_{M,s'})$. For non-interacting fermions, $G_M$ is independent of auxiliary field $\{s\}$ and thus this term vanishes. As a result, we find $S_2 = -2\log|X_M^\rho(\frac{\pi}{2})|$ for non-interacting particles as shown in Eq. (8). For interacting particles, in general this relation between $S_2$ and $X_M^\rho(\frac{\pi}{2})$ no longer holds. Below, we study two interacting models (in 1D and 2D respectively) to explore the difference and connection between these two quantities.

### B. Disorder operator and $S_2$ in a Luttinger liquid

Let us study disorder operator in a 1D spinless LL [63]. The Hamiltonian can be written as

$$H_L = \frac{v_F}{2\pi} \int dx \left[ K(\partial_x \vartheta)^2 + K^{-1}(\partial_x \phi)^2 \right]. \qquad (23)$$

Here $v_F$ is the Fermi velocity, $K$ is the Luttinger parameter.

In models with SU(2) spin rotation symmetry (e.g. the Hubbard model), the Luttinger parameter $K_\sigma = 1$ to preserve the SU(2) symmetry. With the bosonization formula ( See Sec. V in SM [56] ), we can easily compute two channels of the

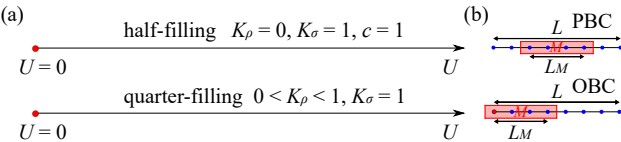

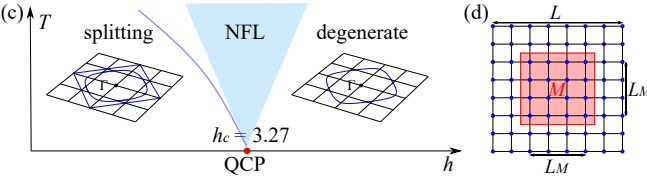

FIG. 3. (a) The zero temperature phase diagram of Hubbard chain at half- and quarter- fillings. The Luttinger parameters and conformal central charge are shown. (b) The entangling subregion $M$ (red) in the system with PBC and OBC, where $M$ contains $L_M$ lattice sites. (c) The $T$-$h$ phase diagram of the 2D interacting fermion model, where fermions couple to ferromagnetic Ising spins with a transverse field $h$. The nFL region, where it was shown the quantum part of the fermionic self-energy satisfies $\sim \omega_n^{2/3}$ [51], lies above the QCP at $h_c = 3.27$. The paramagnetic(ferromagnetic) phase is denoted by degenerate(splitting) FSs, and both of which is classified as FL behavior.

(d) The entangling region $M$ (red) defined on a $L \times L$ square lattice with PBC, where $M$ contains $L_M \times L_M$ sites.

disorder operators,

$$-\log|X_M^\rho(\theta)| = \frac{\theta^2 K_\rho}{\pi^2}\log L_M + \cdots$$
$$-\log|X_M^\sigma(\theta)| = \frac{\theta^2 K_\sigma}{4\pi^2}\log L_M + \cdots \quad (24)$$

In a 1D LL, the 2nd Rényi entropy $S_2$ is given by $S_2 = \frac{c}{4}\log L_M + \cdots$. Notice that the coefficient of the logarithmic term does not depend on Luttinger parameters at all. On the other hand, as we have just shown the disorder operators, while having similar logarithmic scaling with $L_M$, strongly depend on Luttinger parameters and eventually interactions (especially $X_M^\rho$).

To demonstrate this difference, here we consider a 1D repulsive Hubbard chain

$$H_U = -t\sum_{\langle ij\rangle,\sigma}\hat{c}_{i\sigma}^\dagger\hat{c}_{j\sigma} + U\sum_i \hat{n}_{i\uparrow}\hat{n}_{i\downarrow}. \quad (25)$$

At half-filling, the ground state of this model has a charge gap ($\Delta_c \sim e^{-t/U}$ when $U$ small), while the spin degrees of freedom remain gapless [64]. For the charge disorder operator, since $K_\rho = 0$ we expect $-\log|X_M^\rho(\theta)|$ to be a constant without logarithmic correction. Whereas for the spin disorder operator we should have $-\log|X_M^\sigma(\theta)| = \frac{\theta^2}{4\pi^2}\log L_M + \text{const.}$. For EE (Rényi entropy $S_2$), we expect $S_2 = \frac{c}{4}\log L_M + \text{const.}$, with $c = 1$ due to the gapless spin channel.

We compute the disorder operator with $\theta = \frac{\pi}{2}$ and EE in the $L = 64$ chains with PBC to the high precision via DMRG and DQMC simulations. As shown in Fig. 4(a), at $U = 3$, the disorder operator and EE behave as expected, that is, the

disorder operator becomes a plateau in the bulk as $K_\rho = 0$, and EE show a clear $\log L_M$ dome-like behavior due to $c = 1$. In Fig. 4(b), we further verify this by plotting EE and disorder operator versus the conformal distance $\tilde{L}_M$, and the slopes of the darker blue and red data give central charge $c \simeq 1.009$ and $K_\rho \simeq 0.004$, well consistent with our expectation.

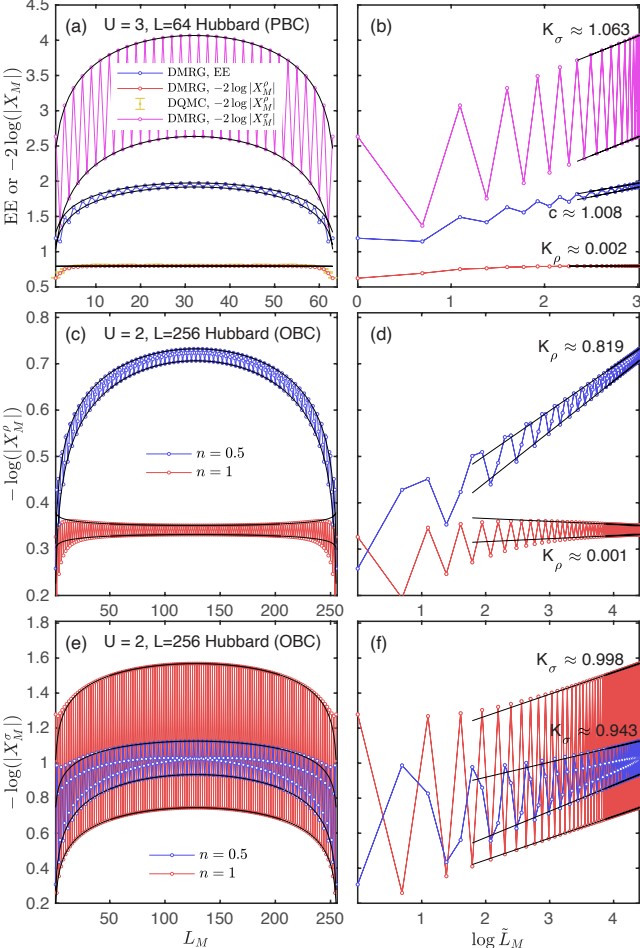

FIG. 4. (a) Rényi entropy $S_2$ and disorder operators, $-2\log|X_M^\rho(\frac{\pi}{2})|$ and $-2\log|X_M^\sigma(\pi)|$, for a half-filled Hubbard chain with PBC at $U = 3$ and system size $L = 64$. The data points are obtained via DMRG and DQMC, and the horizontal axis $L_M$ is the size of the region $M$. Figure (b) shows the same data vs the log of the conformal distance $\log\tilde{L}_M = \log(\frac{L}{\pi}\sin\frac{\pi L_M}{L})$, which should be fitted to a linear function at large $\log\tilde{L}_M$, and we find that $c \simeq 1.009$, $K_\rho \simeq 0.004$ and $K_\rho \simeq 1.063$. (c,d) Charge disorder operator $-\log|X_M^\rho(\frac{\pi}{2})|$ in a Hubbard chain with OBC at half- or quarter- fillings. Here we utilize DMRG to compute the disorder operator and set $U = 2$ and $L = 256$. Fitting in Fig. (d) gives $K_\rho \simeq 0.001$ (half filling) and $K_\rho \simeq 0.819$ (quarter filling), consistent with the Bethe ansatz results $K_\rho = 0$ (half filling) and $K_\rho \simeq 0.82$ (quarter filling) [65]. (e,f) Spin disorder operator $-\log|X^\sigma(\pi)|$ for the same model. The fitting gives $K_\sigma \simeq 0.998$ for half filling and $K_\sigma \simeq 0.943$ for quarter filling, consistent with the expected value $K_\sigma = 1$.

Our new discovery beyond preivous knowledge is that, we find the disorder operator serves as a highly efficient tool

for extracting Luttinger parameters. Traditionally, in DMRG simulations, the $K_\alpha$ is determined from the structure factor $S_\alpha(k) \sim \frac{K_\alpha}{\pi}|k|$ at $k \to 0$, with $\alpha$ being $\rho$ or $\sigma$, representing the charge or spin sector. For systems with a small gap, this approach can suffer from serious finite-size effects, making it very challenging to obtain the correct values of $K_\alpha$. Specifically, for the half-filled Hubbard chain studied here, because the charge gap closes exponentially as $U$ decreases towards zero, an exponentially large system size $L$ is required to overcome the finite-size effect to obtain $K_\rho$. For example, as shown in Ref. [66], at $U = 1.6$, the conventional approach gives a value of $K_\rho \sim 0.5$ at half-filling, even with system size as large as $L = 512$ (See also Sec. VI in SM [56]). In contrast, if we fit the disorder operator with its scaling form $-\log|X_M^\rho(\frac{\pi}{2})| = \frac{K_\rho}{4}\log L_M + \text{const.}$, the finite-size effect is overcome with ease. As shown in Fig. 4 (c,d), with $L = 256$, this fitting is already sufficient to provide the value of $K_\rho$ with high accuracy at and away from half filling. From this fitting, we get $K_\rho \simeq 0.001$ and $K_\rho \simeq 0.819$ at half and quarter fillings ($n = 1$ and $n = 0.5$) respectively, in perfect agreement with exact results from the Bethe ansatz: $K_\rho = 0$ at half-filling and $K_\rho \simeq 0.82$ at quarter filling [65]. In Fig. 4 (e,f), we also check the Luttinger parameter of the spin sector $K_\sigma$ for both half-filling ($n = 1$) and quarter-filling ($n = 0.5$), and find $K_\sigma \simeq 0.998$ for $n = 1$ and $K_\sigma \simeq 0.943$ for $n = 0.5$, in good agreement with the expected value $K_\sigma = 1$.

## C.  2D Fermi Liquid and non-Fermi Liquid

We now move on to study the disorder operator and the EE for interacting itinerant fermions in 2D. In contrast to 1D or the non-interacting limits, where precise knowledge about the disorder operator and EE can be obtained from exact solutions and/or effective field theory, 2D itinerant fermions is a much more challenging problem with limited analytical results. In this section, we use DQMC to compute the charge- and spin-disorder operators and the second Rényi entropy, and using the numerical results to examine the connection and difference between disorder operators and the EE.

Both in theory and in DQMC simulations, a very fruitful approach to access FLs and the nFLs is to couple free fermions with critical bosonic fluctuations [49–51, 67–84]. Away from the quantum critical regime, these models provide a FL phase. In the vicinity of the QCP, critical fluctuations drive the system into a nFL phase with over-damped low-energy fermionic excitations [53].

In this study, we utilize one of simplest and well-studied models of this type: spin-1/2 fermions coupled to a ferromagnetic transverse-field Ising model as studied in Refs. [49–53]. The Hamiltonian contains three parts

$$H_{\text{FS}} = H_{\text{f}} + H_{\text{Ising}} + H_{\text{int}}, \qquad (26)$$

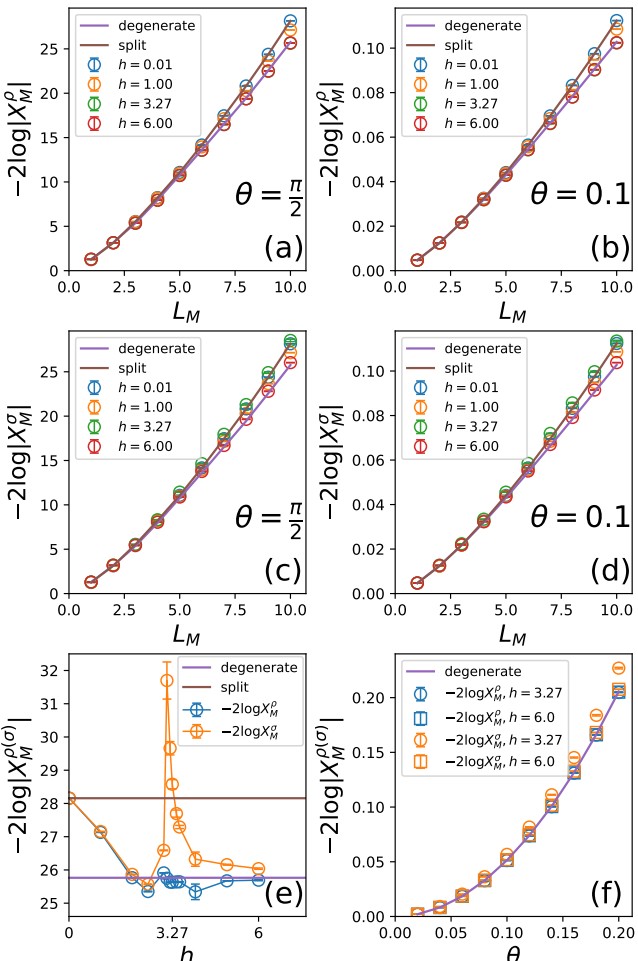

FIG. 5. The charge (a-b) and spin (c-d) disorder operators at various $h$ with $L = 20$ and $\beta = 100$. $h_c = 3.27$ is the QCP, and $h > h_c$ ($h < h_c$) is the paramagnetic (ferromagnetic) phase. The purple (brown) solid line marks the exact formula at the noninteracting limit $h = \infty$ ($h = 0$). From (a)-(d), we find that both $-2\log|X_M^\rho|$ and $-2\log|X_M^\sigma|$ scale as $L_M \log L_M$. For the charge disorder operator $-2\log|X_M^\rho|$, the coefficient of the $L_M \log L_M$ term is very close to the free-fermion prediction [Eq (15)]. In the paramagnetic phase and at the critical point, the FS remain degenerate, and thus $-2\log|X_M^\rho|$ remains almost a constant independent of $h$. In the ferromagnetic phase, because the splitting between the spin up and down FSs increases as we decrease $h$, this change in the FS shape increases the value of $-2\log|X_M^\rho|$. This trend is clearer in Fig. (e), which shows the $h$ dependence of the disorder operator. As for the spin sector, the value of $-2\log|X_M^\sigma|$ shows a sharp peak at the QCP, due to the divergent spin fluctuations, as shown in the main text. In Fig. (f), we plot the $\theta$-dependence of disorder operator. Same as in free fermion systems, we find that both $-2\log|X_M^\rho|$ and $-2\log|X_M^\sigma|$ scales as $\theta^2$, for both the FL phase and at the QCP.

where

$$H_f = -t_1 \sum_{\langle ij \rangle \sigma \lambda} (\hat{c}^\dagger_{i\sigma\lambda} \hat{c}_{j\sigma\lambda} + \text{h.c.}) - \mu \sum_{i\sigma\lambda} \hat{n}_{i\sigma\lambda},$$

$$H_{\text{Ising}} = -J \sum_{\langle ij \rangle} \hat{s}^z_i \hat{s}^z_j - h \sum_i \hat{s}^x_i, \qquad (27)$$

$$H_{\text{int}} = -\frac{\xi}{2} \sum_i \hat{s}^z_i \left( \hat{S}^z_{i,1} + \hat{S}^z_{i,2} \right).$$

The fermionic part $H_f$ consists of two identical layers of fermions labeled by the layer index $\lambda = 1, 2$, and $\sigma = \uparrow, \downarrow$ labels the fermion spin. Here, fermions can hop between neighboring sites of a square lattice ($t_1$) and $\mu$ is the chemical potential. The bosonic part $H_{\text{Ising}}$ describes quantum Ising spins with ferromagnetic interactions subject to a transverse field $h$. And the Ising spins live on the same square lattice as fermions. In the absence of fermions, these Ising spins form a paramagnetic (ferromagnetic) phase if $h > h_c$ ($h < h_c$), separated by a QCP, which belongs to the 2+1D Ising universality class at $h = h_c \approx 3.04$. The last term $H_{\text{int}}$ couples the fermion spins with Ising spins at the same lattice sites. With this coupling, the paramagnetic-ferromagnetic phase transition for the Ising spins now induces a quantum phase transition for the fermions, i.e., a paramagnetic-ferromagnetic phase transition for itinerant fermions. As shown in Refs. [49–51], $h > h_c$ the system is in the paramagnetic phase where spin up and down fermions are degenerate and share the same FS. For $h < h_c$, the model has an itinerant ferromagnetic phase, where spin-up and down FSs splits, due to the spontaneous magnetization of Ising spins which effectively provide opposite chemical potentials for fermions with opposite spin flavors. Away from the QCP, the fermions form a FL with well-defined quasiparticles, but have different shape of FSs in paramagnetic and ferromagnetic phase. The $h \to 0$ and the $h \to \infty$ cases are regarded as classical ordered and decoupled limits, respectively. $h = h_c$ is the QCP that separate these two phases, where critical fluctuations destroy the coherence of fermionic particles where fermionic excitations become over-damped, and the FS is smeared out and result in a nFL phase with fermion self-energy scales as $\sim \omega_n^{2/3}$ [51].

As shown in Ref. [49], the global internal symmetry of $H_{\text{FS}}$ is identified as

$$\text{SU(2)}_\uparrow \times \text{SU(2)}_\downarrow \times \text{U(1)}_\uparrow \times \text{U(1)}_\downarrow \times \mathbb{Z}_2, \qquad (28)$$

where the $\text{SU(2)}_\uparrow \times \text{SU(2)}_\downarrow$ group consists of independent rotations in the layer $\lambda$ basis for each spin species of fermions. The $\text{U(1)}_\uparrow \times \text{U(1)}_\downarrow$ symmetries correspond to conservation of particle number of spin up and spin down fermions, and the $\mathbb{Z}_2$ symmetry, generated by $\prod_i \hat{s}^x_i \prod_i e^{i\pi(S^x_{i,1} + S^x_{i,2})}$, acts as $S^z_{i,\lambda} \to -S^z_{i,\lambda}, s^z_i \to -s^z_i$, where the latter is the symmetry breaking channel and defines the order parameter of the ferromagnetic phase.

In addition, this Hamiltonian is invariant under the antiunitary symmetry $i\tau_y K$, where $\tau_y$ is a Pauli matrix in the layer basis and $K$ is the complex conjugation operator. Thanks to this symmetry, the QMC simulation is free of sign problem [85].

From these symmetries, multiple disorder operators can be introduced: for example, the charge disorder operator shown in Eq. (2) from the U(1) charge conservation and the spin disorder operator defined in Eq. (3) due to the conservation of the $z$ component of the fermion spin $S_z$. We note the Ising symmetry of $H_{\text{FS}}$ in Eq. (26) also enables us to define this disorder operator $\hat{S}^z_M = \prod_{i \in M} s^x_i e^{i\pi(S^x_{i,1} + S^x_{i,2})}$. In this study, however, we will focus on the charge and spin disorder operators, Eqs. (2) and (3), while other disorder operators will not be considered due to the technical challenges to measure them in DQMC. The reason lies in the fact that, the DQMC simulations are performed on the Ising spins $s^z$ and fermion spins $S^z$ basis, rendering the measurement of $x$ component of these spins costly. While for the charge density and $z$ component of the spin operator, large system sizes up to $L \times L = 24 \times 24$ and low temperatures down to $\beta = 100$ can be accessed.

We note, in principle, the EE is not directly related to any global symmetry of the system, and as we will show below, it can reveal not only the $L_M \log L_M$ scaling behavior coming from the geometry of FS in an interacting FL, but also the quantum critical scaling at QCP whose precise scaling form is still unknown. Here we see clearly that the disorder operator can indeed capture the $L_M \log L_M$ universal entanglement scaling in the interacting fermion models as that of the EE, but at the same time, extra symmetry consideration is needed, if one is asking for the particular information in the QCP entanglement.

Fig. 5(a) shows $X^\rho_M(\frac{\pi}{2})$ for $L = 20$. The model has two exactly solvable limits, $h \to \infty$ and $h \to 0$, where fermions are non-interacting, i.e., the system is a non-interacting Fermi gas. At $h \to \infty$, spin up and down fermions share the same FS, while at $h = 0$, the spin-up and down FSs split, due to the ferromagnetic order. For both these two limits, the exact solutions indicate that $-\log |X^{\rho/\sigma}_M(\theta)| \sim s(\theta) L_M \log L_M$ with $s(\theta) \propto \theta^2$ and the proportionality coefficient is determined by the shapes of the region $M$ and the FS, as shown in Eq. (15). These two exact solvable limits are shown in Fig. 5 as solid lines marked as "degenerate" ($h \to \infty$) and "split" ($h = 0$) respectively.

As shown in Fig. 5(a) and (b), for the paramagnetic phase ($h > h_c$), the charge disorder operator depends very weakly on $h$, and its value remains nearly the same as the non-interacting limit $h \to \infty$, even if $h$ reaches the critical value $h_c$. This is very different from the 1D case, where the value of the disorder operator starts to change immediately even if an infinitesimal amount of interactions are introduced, due to the change in Luttinger parameters. This observation, if taken at face value, seems to indicate that interactions between fermions is irrelevant for $X^\rho_M$, and the non-interacting formula [Eq. (15)] survives in the FL phase. However, as will be discussed below, a more careful analysis indicates the opposite. For the ferromagnetic phase ($h < h_c$), the numerical data indicates that $-\log |X^\rho_M(\theta)|$ still scales as $\theta^2 L_M \log L_M$, but with coefficient that gradually increases as we decrease $h$. This is again largely consistent with the non-interacting formula Eq. (15). Because the FS splits in the ferromagnetic phase and this splitting increases when $h$ reduces towards zero, the coefficient of the $L_M \log L_M$ term should shift ac-

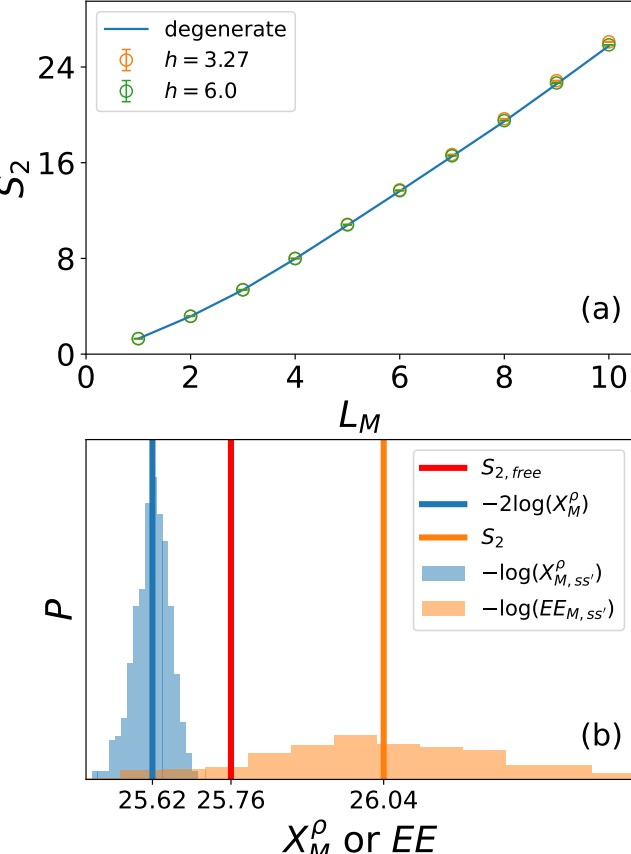

FIG. 6. (a) $S_2$ versus $L_M$ at $h = h_c = 3.27$ and $h = 6.0$. The value is close to the free-fermion limit (solid line), but with some small deviations, which will be analyzed in panel (b). (b) Distribution of $EE_{M,ss'}$ and $X^\rho_{M,ss'}$ over different auxiliary-field configurations. The mean values of these two quantities, after taken a log, are $S_2$ and $-2\log|X^\rho_M(\frac{\pi}{2})|$, which are marked by the orange and blue vertical lines respectively. This distribution demonstrate the error bar of $S_2$ and $-2\log|X^\rho_M(\frac{\pi}{2})|$, and it is clear that $S_2$ ($-2\log|X^\rho_M(\frac{\pi}{2})|$) is larger (smaller) that the free-fermion limit, $S_{2,free}$ marked by the vertical red line. In contrast to free systems, where $S_2$ and $-2\log|X^\rho_M(\frac{\pi}{2})|$ coincide, at QCP, we find that these two quantities clearly deviates from each other, and both of them differ from their free theory value.

cording to the shape of the FS. This trend of $X^\rho_M$ is summarized in Fig. 5(e).

As for the spin disorder operator, we plotted the two noninteracting limits, $h \to \infty$ and $h \to 0$, in Fig. 5(c) and (d) as the solid lines. In the non-interacting limit, $|X^\sigma_M(\theta)|$ obeys the exact identity $|X^\sigma_M(\theta)| = |X^\rho_M(\frac{\theta}{2})|$, and thus it doesn't carry any extra information beyond the charge disorder operator. For interacting fermions $0 < h < \infty$, we find that $|X^\sigma_M(\theta)|$ starts to deviate from $|X^\rho_M(\frac{\theta}{2})|$. Away from the QCP, the deviation is small. However, near the QCP a much larger deviation is observed and $-\log|X^\sigma_M(\theta)|$ shows a peak at $h \sim h_c$, as shown in Fig. 5(e). This peak of $-\log|X^\sigma_M(\theta)|$ is due to the fact that the spin disorder operator happens to be the order parameter of this QCP. As shown early on, at small $\theta$,

$-\log|X^\sigma_M(\theta)|$ measures the fluctuations of $\sigma_z$ in the region $M$, i.e., $-\log|X^\sigma_M(\theta)|/\theta^2 = \langle(\hat{S}^z_M - \langle\hat{S}^z_M\rangle)^2\rangle$, which develops a peak at the QCP. The location of this peak marks the critical value of $h$, at which spin fluctuations are pronounced and thus can be used as a tool to detect the QCP.

In addition to the disorder operators, we also compute the second Rényi entropy $S_2$. Previous variational Monte Carlo studies of EE for trial wavefunctions of both FL, composite FL and spinon FS states [86–89] suggest that they all obey the $L_M \log L_M$ scaling. We calculate $S_2$ utilizing Eq. (22) via joint distributions. In Fig. 6(a), the solid line is the exact formula for Rényi entropy at the non-interacting limit $h \to \infty$, and the dots are numerically measured $S_2$ in the paramagnetic phase $h = 6.0$ and at the QCP $h = h_c = 3.27$. Naively, this figure seems to indicate that the value of $S_2$ is independent of $h$ in the entire paramagnetic phase and at the QCP. However, as will be shown below, more careful analysis indicates the opposite conclusion.

Before we discuss a more careful analysis in the next section, let us conclude this section by providing a quick summary of Figs. 5 and 6. Using DQMC simulations, we find that the spin disorder operator is very sensitive to this itinerant Ising QCP, and exhibits a diverging peak, because it coincides with the order parameter of this QCP and thus direct probes the diverging critical fluctuations. As for the charge disorder operator and $S_2$, they seem to behave as the non-interacting limit. If this conclusion is true, it implies that the exact relation of the non-interacting limit, $S_2 = -2\log|X^\rho_M(\frac{\pi}{2})|$, should survive in the FL. Because measuring $|X^\rho_M(\frac{\pi}{2})|$ costs much less numerical resource with much better data quality (equal time measurement without replicas) than $S_2$, this observation seems to provide a much more efficient methods to probe $S_2$ in FLs and nFLs (near QCPs). However, as will be shown in the next section, more careful analysis indicates that $|X^\rho_M(\frac{\pi}{2})|$ and $S_2$ are clearly distinct in interacting systems. The difference between them is small (but nonzero) in the FL phase, and becomes much severer as we approach the QCP and the nFL phase.

### D. Scaling analysis for EE and disorder operators at QCP

To examine the interaction dependence of $S_2$ and $X^\rho_M$, we create $50 \times 50$ QMC auxiliary-field configurations to imitate the joint distribution of $\{s, s'\}$. For each pair of auxiliary-field configuration, we calculate $EE_{M,ss'} = \det[G_{M,s}G_{M,s'} + (\mathbb{1} - G_{M,s})(\mathbb{1} - G_{M,s'})]$ and $X^\rho_{M,ss'} = \det[G_{M,s}G_{M,s'} + (\mathbb{1} - G_{M,s})(\mathbb{1} - G_{M,s'}) + i(G_{M,s} - G_{M,s'})]$ and plot their distributions in Fig. 6(b). Here we set $h = h_c = 3.27$ and 10 independent Markov chains are used to yield the errorbar of each $EE_{M,ss'}$ and $X^\rho_{M,ss'}$ data points. As shown in Eqs. (21) and (22), the mean values of $EE_{M,ss'}$ and $X^\rho_{M,ss'}$, after averaging over all joint auxiliary-field configurations, are $e^{S_2}$ and $X^\rho_M(\frac{\pi}{2})$ respectively.

The logarithms of the two average values are marked in Fig. 6(b) as the orange and blue vertical lines, respectively,

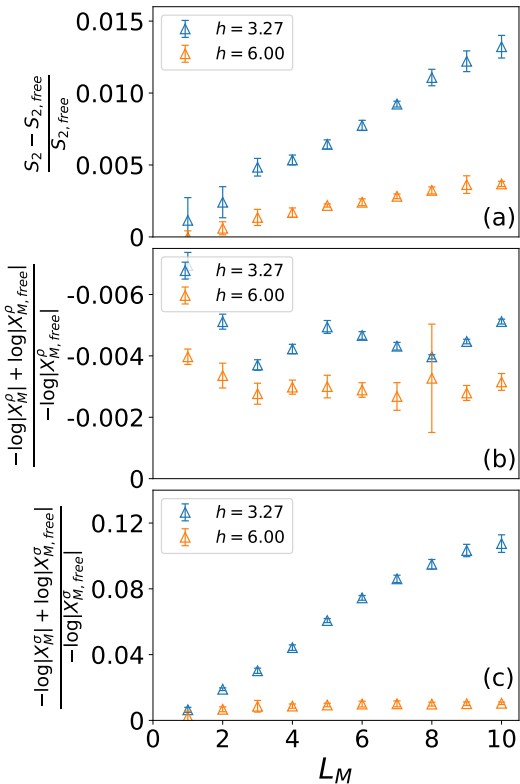

FIG. 7. (a,b,c) Deviation from the non-interacting limit in the FL phase $h = 6.00$ and at the QCP $h = 3.27$ for (a) $S_2$ (b) $-2\log X_M^\rho(\frac{\pi}{2})$ and (c) $-2\log X_M^\sigma(\pi)$. Within the system sizes that we can access, these deviations does not decrease with system size. Instead, the deviation of $S_2$ and $-2\log X_M^\sigma(\pi)$ show clear increase, indicating that it is not due to finite-size effects.

while the red vertical line marks the free-fermion value of $S_2 = -2\log|X_M^\rho(\frac{\pi}{2})|$ at the $h \to \infty$ limit. By comparing the three vertical lines in Fig. 6(b), we find that at the QCP, $S_2$ no longer coincides with $-2\log|X_M^\rho(\frac{\pi}{2})|$. In comparison with non-interacting fermions, interactions increase the value of $S_2$ but reduces the value of $-2\log|X_M^\rho(\frac{\pi}{2})|$. Although the deviation from the free-fermion value is small ($< 2\%$), the distribution shown in Fig. 6(b) clearly indicates that this deviation is beyond the error bar.

More importantly, this deviation from the free-fermion limit is not due to finite-size effects, because it increases as the system size increases towards the thermal dynamic limit, as shown in Fig. 7(a) and (b). For Rényi entropy $S_2$, Fig. 7(a) shows that away from the QCP ($h = 6$), deviations from the free-fermion value increases with $L_M$, but the increase seems to saturate when $L_M$ approaches 12. This saturation behavior suggests that in a FL, $S_2$ has the same scaling form as non-interacting systems, $S_2 \sim s_{2D} L_M \log L_M$, but the coefficient of $s_{2D}$ gradually increases as interactions are turned on. In contrast to the free fermion limit, where $s_{2D}$ only depends on the shapes of the FS and $M$, its value is sensitive to the interaction strength in a FL. At the QCP ($h = 3.27$), the deviation of $S_2$ from the free-fermion limit increases much faster than the FL phase [Fig. 7(a)], and up to the largest system size that

we can access $L_M = 12$ and $L = 24$, a saturation is not clearly observed. This increasing trend indicates three possible scenarios. (1) If the increase never saturates at large $L_M$, it would indicate that $S_2$ at the QCP follows a different scaling form, which increases faster than $L_M \log L_M$, e.g., $L_M^\alpha$ with a power between 1 and 2 or $L_M(\log L_M)^\alpha$ with $\alpha > 1$. (2) If the increase eventually saturates at large $L_M$, it would indicate that $S_2$ still follows the functional form of $L_M \log L_M$, but with a coefficient much larger than free-fermions ($h \to \infty$) or the FL phase ($h = 6$). (3) In the third scenario, the deviation eventually starts to decrease at some large $L_M$. If it decrease back to zero at the thermodynamic limit ($L_M \to \infty$), it would imply that interactions and QCPs have no impact on $S_2$, whose value remains identical to the free theory. Although this third scenario is in principle possible, to the largest system size that we can access, we don't observe any signature for this deviation to start decreasing at large $L_M$, and thus no evidence supports it. For both scenarios (1) and (2), critical fluctuations near the QCP have a nontrivial impact on $S_2$, i.e., $S_2$ is sensitive to the presence of an itinerant QCP.

For the charge disorder operator $X_M^\rho(\frac{\pi}{2})$, we find from Fig. 7(b) that the deviation from the free fermion limit seems to saturate at large $L_M$, indicating that $X_M^\rho(\frac{\pi}{2})$ maintains its functional form $\sim L_M \log L_M$ in the FL and nFL phases, and the deviation only modifies that coefficient of this $L_M \log L_M$ term. This deviation seems to increase a bit as we approach the QCP ($h = 3.27$), in comparison with the FL phase $h = 6.00$. However, this increase is much weaker than the increase of the deviation of $S_2$ at the QCP, indicating that the charge disorder operator is much less sensitive to the itinerant QCP than $S_2$.

For comparison, we also plot the deviation for the spin disorder operator from the free fermion limit in Fig. 7(c). Away from the QCP, the deviation seems to saturate. At the QCP, the deviation increases dramatically, indicating that the spin disorder operator is very sensitive to the quantum phase transition. This observation is consistent with the diverging spin fluctuations discussed in the previous section. Similar with the $S_2$ case, whether the $L_M \log L_M$ scaling form will change, or, there is a large coefficient at the QCP, are beyond our current system sizes.

In summary, we find that although the values of $S_2$ and $-2\log|X_M^\rho(\frac{\pi}{2})|$ seems to be close to the free fermion limit, interactions push their values towards different directions, i.e., increasing $S_2$ and decreasing $-2\log|X_M^\rho(\frac{\pi}{2})|$. These two opposite trends are beyond the numerical error bar, and the effects becomes stronger as the system size increases, indicating that it is not due to the finite-size effect. In the FL phase away from the QCP, the difference between $S_2$ and $-2\log|X_M^\rho(\frac{\pi}{2})|$ are small, but it become much severer at the QCP. The spin disorder operator, $-2\log X_M^\sigma(\pi)$, on the other hand, does exhibit enhanced signal at the QCP.

## VI. DISCUSSIONS

Accessing entanglement measures in interacting fermion systems has been a long standing problem. Early attempts

that do not require replicas [11, 12] are often plagued by large fluctuations. Implementing replicas [13, 90] allows one to circumvent these issues but is then numerically demanding. The same holds for the entanglement spectra and Hamiltonian [14]. In this work, we investigate an alternative, namely, the fermion disorder operator which provides similar entanglement information. From the technical point of view, this quantity does not rely on replicas and can be computed on the fly in an auxiliary field QMC simulation.

Besides the computational superiority, the disorder operators also have close relationship with the available observables for experimental measusments, thus plays a role as entanglement witness. In quantum point contact model, the entanglement of such systems is produced by the transmitted charges and measured via the statistics or distribution of charge, where disorder operator in charge channel also writes in this form. Although complicated experimental settings lead to difficulties for obtaining entanglement infomation, the simplicity of disorder operator may give inspiration for designing experimental measurements.

Generically, the disorder operator and Rényi EEs are different quantities. In particular, the disorder operator is formulated in terms of a global symmetry of the model system, whereas the Rényi entropies are defined without any symmetry considerations. At small angles, the disorder operator relates to so-called bi-partite fluctuations that have been introduced as an entanglement witness [10]. Despite the obvious differences, for non-interacting fermionic systems, there is a one-to-one mapping between the *charge* disorder operator at a given angle, and the Rényi entropies. As such, for a bipartition of space with $L_M^d$ the volume of one partition, we observe a $L_M^{d-1} \log L_M$ law for the FL for both quantities.

Beyond the non-interacting limit, notable differences appear. For the 1D case with spin and charge degrees of freedom, the prefactor of the $\log L_M$ law for the spin and charge disorder operators captures the Luttinger parameter in the respective sectors. This contrasts with the Rényi entropy that picks up the central charge. In fact, one of our discovery in this work is that the disorder operator offers a much better estimation of the Luttinger parameter (the coefficient in the $\log L$ scaling) as compared to the traditional fitting from the structure factors.

To investigate the nature of the disorder operator in 2D, we concentrate on metallic Ising ferromagnetic quantum criticality [49, 51]. *Deep* in the ordered and disordered FLs phases, the spin and charge disorder operators show very similar behaviors and follow an $L_M \log L_M$ law with prefactor dictated by the FS topology. In the proximity of the phase transition we observe marked differences between both symmetry sectors. In fact, in this special case, the generator of the U(1) spin symmetry corresponds to the order parameter and the spin-disorder operator shows singular behavior at criticality. On the other hand, the charge disorder operator does not pick up the phase transition and smoothly interpolates between the order and disordered FL phases. Remarkably, and on the considered lattice sizes, the scaling of the spin disorder operator at criticality reflects that of the Rényi entropy, and supports the interpretation of either a deviation from the $L_M \log L_M$

law, or, greatly enhanced coefficient of the scaling form. Our examples in 1D and 2D interacting fermion systems, illustrate the symmetry dependence in the design and interpretation of the disorder operator.

Given that, the present work constitutes a first comprehensive and thorough investigation of the disorder operator for both free and more importantly – interacting fermion systems in 1 and 2D, where the latter has not been considered before. It is in this new direction, the similarities and differences between disorder operator and EE in interacting fermions have been thoroughly investigated in our lattice model calculations in 1D and more importantly 2D, which allow for a more profound understanding of entanglement properties in interacting fermionic systems and can inspire future research in this direction.

A key point in considering the disorder operator is that it seems possible to access experimentally. As mentioned previously, at small angles, it maps onto two-point correlation functions of local operators. Such quantities are routinely computed in scattering experiments. For example neutron scattering experiments provide the the dynamical spin structure factor, from which equal time correlation functions can be extracted. With this in mind, understanding the intricacies of the disorder operator and what we can learn from it for various phases of correlated quantum matter and across various QCPs becomes more pressing. We foresee a number of future investigations with disorder operator that include, possible finite temperature properties and extension to the entanglement of the mixed state [32], on-going experiments in quantum switch device and optical lattices [91–93], lattice models for quantum critical metals [54, 55], correlated insulators and superconductors in moiré lattice models [94–101], exotic states of matter such as deconfined quantum criticality, emergent quantum spin liquids and topological fermionic states [31, 102–111].

## ACKNOWLEDGEMENT

W.L.J., B.-B.C. and Z.Y.M. would like to thank Zheng Yan for insightful discussions on the EE and ES, they acknowledge the support from the Research Grants Council (RGC) of Hong Kong SAR of China (Project Nos. 17301420, 17301721, AoE/P-701/20, 17309822, HKU C7037-22G), the ANR/RGC Joint Research Scheme sponsored by RGC of Hong Kong and French National Research Agency (Project No. A_HKU703/22), the Strategic Priority Research Program of the Chinese Academy of Sciences (Grant No. XDB33000000), the K. C. Wong Education Foundation (Grant No. GJTD-2020-01) and the Seed Fund "Quantum-Inspired explainable-AI" at the HKU-TCL Joint Research Centre for Artificial Intelligence. M.C. acknowledges support from NSF under award number DMR-1846109. We thank the HPC2021 platform under the Information Technology Services at the University of Hong Kong, and the Tianhe-II platform at the National Supercomputer Center in Guangzhou for their technical support and generous allocation of CPU time. F.F.A. and Z.L thank Cenke Xu and Chaoming Jian for discussion on the relation of the disorder operator and Rényi

entropies for free electrons. F.F.A. acknowledges support from the DFG funded SFB 1170 on Topological and Correlated Electronics at Surfaces and Interfaces. Z.L. thanks the Würzburg-Dresden Cluster of Excellence on Complexity and Topology in Quantum Matter ct.qmat (EXC 2147, project-id 390858490) for financial support.

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

## VII. SUPPLEMENTAL MATERIALS FOR

## MANY VERSUS ONE: THE DISORDER OPERATOR AND ENTANGLEMENT ENTROPY IN FERMIONIC QUANTUM MATTER

### A. Section I: Disorder operator in QMC calculation

#### 1. Disorder operator in charge channel

We explain the implementation of disorder operator in the determinant QMC. We first consider the disorder operator in charge channel, $\hat{X}_M^\rho(\theta) = \prod_{i \in M} e^{i\hat{n}_i\theta}$. For convenience, we denote $X_M^\rho = \langle \hat{X}_M^\rho(\theta) \rangle$. Below, we show that in QMC simulation, the expectation value of the disorder operator is an equal-time measurement of fermion Green's function.

Define the fermion Green's function matrix of certain configuration, $G_{s,i\sigma,j\sigma'} = \langle \hat{c}_{i\sigma}\hat{c}_{j\sigma'}^\dagger \rangle_s$. $i, j$ labels the lattice site, and $s$ labels the configuration. One has,

$$X_M^\rho = \sum_s P_s \frac{\text{Tr}[\hat{U}_s(\beta,\tau)\prod_{i\in M} e^{i\hat{n}_i\theta}\hat{U}_s(\tau,0)]}{\text{Tr}[\hat{U}_s(\beta,0)]} \tag{29}$$

where $P_s = \frac{\text{Tr}[\hat{U}_s(\beta,0)]}{\sum_s \text{Tr}[\hat{U}_s(\beta,0)]}$ is the normalized weight of each configuration. We define $N_M(N_{M,\sigma})$ to be the number of sites(spin flavor) of whole system, $N_\sigma$ is the number of spin flavors. The total Hamiltonian $\hat{H}$ can be expressed in site and spin basis, and constructed as a matrix $H$ with dimension $N_\sigma N \times N_\sigma N$. Applying the expression $\text{Tr}[e^{\hat{H}}] = \text{Det}[1 + e^H]$, and $\text{Det}[1 + AB] = \text{Det}[1 + BA]$,

$$\begin{aligned}
X_M^\rho &= \sum_s P_s \frac{\text{Det}[\mathbb{1} + B_s(\beta,\tau)e^{T^\rho}B_s(\tau,0)]}{\text{Det}[\mathbb{1} + B_s(\beta,0)]} \\
&= \sum_s P_s \frac{\text{Det}[\mathbb{1} + B_s(\tau,0)B_s(\beta,\tau)e^{T^\rho}]}{\text{Det}[\mathbb{1} + B_s(\tau,0)B_s(\beta,\tau)]} \\
&= \sum_s P_s \frac{\text{Det}[\mathbb{1} + (G_s^{-1} - \mathbb{1})e^{T^\rho}]}{\text{Det}[G_s^{-1}]} \\
&= \sum_s P_s \text{Det}[G_s + (\mathbb{1} - G_s)e^{T^\rho}] \\
&= \sum_s P_s X_{M,s}^\rho
\end{aligned} \tag{30}$$

$T^\rho$ is the $N_\sigma N \times N_\sigma N$ diagonal matrix with $T_{i\sigma,i\sigma}^\rho = \begin{cases} i\theta, i \in M \\ 0, i \notin M \end{cases}$ .

We first swap the index of $G$ to seperate the sites in $M$, and out of $M$, which does not change the determinant. We denote $N_M$ as the number of sites in region $M$ and site $i \in M$, $j \notin M$. Then the diagonal element of $e^T$ transfers to $[\underbrace{e^{i\theta}, \cdots, e^{i\theta}}_{N_\sigma N_M}, \underbrace{1, \cdots, 1}_{N_\sigma(N-N_M)}]$. Then, we have

$$\begin{aligned}
X_{M,s}^\rho &= \text{Det}\left[ \begin{pmatrix} G_{ii} & \cdots & G_{ij} \\ \vdots & \ddots & \vdots \\ G_{ji} & \cdots & G_{jj} \end{pmatrix}_{M,s} + \begin{pmatrix} 1-G_{ii} & \cdots & -G_{ij} \\ \vdots & \ddots & \vdots \\ -G_{ji} & \cdots & 1-G_{jj} \end{pmatrix}_{M,s} \begin{pmatrix} e^{i\theta} & \cdots & 0 \\ \vdots & \ddots & \vdots \\ 0 & \cdots & 1 \end{pmatrix} \right] \\
&= \text{Det}\left[ \begin{pmatrix} G_{ii} & \cdots & G_{ij} \\ \vdots & \ddots & \vdots \\ G_{ji} & \cdots & G_{jj} \end{pmatrix}_{M,s} + \begin{pmatrix} (1-G_{ii})e^{i\theta} & \cdots & -G_{ij} \\ \vdots & \ddots & \vdots \\ -G_{ji}e^{i\theta} & \cdots & 1-G_{jj} \end{pmatrix}_{M,s} \right] \\
&= \text{Det}\left[ \begin{pmatrix} G_{ii} + (1-G_{ii})e^{i\theta} & \cdots & 0 \\ \vdots & \ddots & \vdots \\ G_{ji}(1-e^{i\theta}) & \cdots & 1 \end{pmatrix}_{M,s} \right] \\
&= \text{Det}\left[ e^{i\theta}\mathbb{1} + (1-e^{i\theta})G_{M,s} \right]
\end{aligned} \tag{31}$$

where $G_{M,s}$ represents the Green's function matrix projection on $M$ with $N_\sigma N_M \times N_\sigma N_M$ dimension, i.e., $G_{M,s,i\sigma,j\sigma'} = G_{s,i\sigma,j\sigma'}$ for $i,j \in M$.

Furthermore, if $s_z$ is the conserved quantity, i.e. $G_M$ is block-diagonal and $G_{s,i\sigma,j\sigma'} \propto \delta_{\sigma\sigma'}$. Here, $X_{M,\sigma,s}$ denotes the projection of $X_{M,s}$ on the spin basis $\sigma$. And Eq. (31) simplifies to,

$$
\begin{aligned}
X_{M,s}^\rho &= \prod_\sigma \mathrm{Det}\left[e^{i\theta} + (\mathbb{1} - e^{i\theta})G_{M,\sigma,s}\right] \\
&= \prod_\sigma X_{M,\sigma,s}^\rho
\end{aligned}
\tag{32}
$$

### 2. Disorder operator in spin channel

Likewise in above subsection, we consider the disorder operator in spin channel, $\hat{X}_M^\sigma(\theta) = \prod_{i\in M} e^{i\hat{S}_i^z\theta}$ where $\hat{S}_i^z = \frac{1}{2}(\hat{n}_{i\uparrow} - \hat{n}_{i\downarrow})$ is the $z$-direction spin operator for fermion at lattice site $i$, and use $X_M^\sigma = \langle \hat{X}_M^\sigma(\theta) \rangle$. The difference between the disorder operator of spin and charge channel comes from the matrix $T$, where $T_{i\sigma,i\sigma}^\sigma = \frac{i\theta}{2}$ when $i \in M, \sigma = \uparrow$, $T_{i\sigma,i\sigma}^\sigma = \frac{-i\theta}{2}$ when $i \in M, \sigma = \downarrow$ and $T_{i\sigma,i\sigma}^\sigma = 0$ for $i \notin M$.

Utilizing the same derivation as above and given $s_z$ is the conserved quantity, eventually we have,

$$
\begin{aligned}
X_{M,s}^\sigma &= \mathrm{Det}\left[e^{\frac{i\theta}{2}} + (\mathbb{1} - e^{\frac{i\theta}{2}})G_{M,\uparrow,s}\right] \times \mathrm{Det}\left[e^{\frac{-i\theta}{2}} + (\mathbb{1} - e^{\frac{-i\theta}{2}})G_{M,\downarrow,s}\right] \\
&= X_{M,\uparrow,s}^\sigma X_{M,\downarrow,s}^\sigma
\end{aligned}
\tag{33}
$$

We notice that the relation between two channels for certain auxiliary field $s$, $X_{M,\uparrow,s}^\rho(\theta) = X_{M,\uparrow,s}^\sigma(2\theta)$ and $= X_{M,\downarrow,s}^\rho(\theta) = X_{M,\downarrow,s}^{\sigma,*}(2\theta)$. If $Z_2$ symmetry of inversed spin is not broken, e.g. the disorder phase of Fig. 3(c) in the main text, one additionally has $X_{M,\uparrow,s}^\rho(\theta) = X_{M,\downarrow,s}^\rho(\theta)$. Following, we denote $X_{M,\sigma,s}(\theta) = \mathrm{Det}\left[e^{i\theta} + (1 - e^{i\theta})G_{M,\sigma,s,ii}\right]$ to simplify the derivation.

## B. Section II: Disorder operator at various angle

### 1. Small angle expansion

First, we derive the small angle expansion of the disorder operator in continuum limit as a general consideration. Take charge channel as the example,

$$
\begin{aligned}
-\log|X_M^\rho| &= -\log\left|\left\langle e^{i\theta\hat{N}_M} \right\rangle\right| \\
&= -\log\left|1 + i\theta\left\langle \hat{N}_M \right\rangle - \frac{\theta^2}{2}\left\langle \hat{N}_M^2 \right\rangle + O(\theta^2)\right| \\
&= -\log\sqrt{\left(1 - \frac{\theta^2}{2}\left\langle \hat{N}_M^2 \right\rangle\right)^2 + \theta^2\left\langle \hat{N}_M \right\rangle^2 + O(\theta^2)} \\
&= -\log\sqrt{1 - \theta^2(\langle \hat{N}_M^2 \rangle - \langle \hat{N}_M \rangle^2) + O(\theta^2)} \\
&= \frac{\theta^2}{2}(\langle \hat{N}_M^2 \rangle - \langle \hat{N}_M \rangle^2) + O(\theta^2)
\end{aligned}
\tag{34}
$$

We find the leading order is of $\theta^2$, where the coefficient is the known as the density fluctuations [26, 30], and also has the same definition of the cumulant $C_M^2$.

To further apply Eq. (34) on the lattice, we start from free fermion where there is no auxiliary field $s$ to sample and the $P_s$ reduces to identity matrix.

$$
-\log|X_M^\rho| = -\log\left|\sum_s P_s \prod_\sigma X_{M,\sigma,s}^\rho\right| = -\sum_\sigma \log|X_{M,\sigma}^\rho|
\tag{35}
$$

Notice that diagonal matrix element of $X_M^\rho$ is of order $O(1)$, while for off-diagonal matrix element is of $O(\theta)$. We expand

the determinant to $O(\theta^2)$ term,

$$
\begin{aligned}
X_{M,\sigma}^{\rho} &= \prod_i \left( e^{i\theta} + (1 - e^{i\theta}) G_{M,\sigma,ii} \right) - \sum_{\{i,j\}} \frac{(1 - e^{i\theta})^2}{2} G_{M,\sigma,ij} G_{M,ji} + O(\theta^2) \\
&= \prod_i \left( 1 + i\theta - \frac{\theta^2}{2} + (-i\theta + \frac{\theta^2}{2}) G_{M,\sigma,ii} \right) + \sum_{\{i,j\}} \frac{\theta^2}{2} G_{M,\sigma,ij} G_{M,\sigma,ji} + O(\theta^2) \\
&= 1 + i\theta \sum_i (1 - G_{M,\sigma,ii}) - \frac{\theta^2}{2} \sum_i (1 - G_{M,\sigma,ii}) - \frac{\theta^2}{2} \sum_{\{i,j\}} (1 - G_{M,\sigma,ii})(1 - G_{M,\sigma,jj}) - \frac{\theta^2}{2} C_{M,\sigma}^{2,latt} + O(\theta^2)
\end{aligned}
\tag{36}
$$

where $\{i,j\}$ denotes all different combinations for $i,j \in M$ with $i \neq j$. We define,

$$
C_{M,\sigma}^{2,latt} \equiv - \sum_{\{i,j\}} G_{M,\sigma,ij} G_{M,\sigma,ji} = \sum_{\{i,j\}} \left( \langle \hat{n}_i \hat{n}_j \rangle - \langle \hat{n}_i \rangle \langle \hat{n}_j \rangle \right) \equiv \sum_{\{i,j\}} C_{ij}
\tag{37}
$$

Above, we utilize the Wick theorem. The definition of $C_{M,\sigma}^{2,latt}$ is density fluctuations ( or the cumulant ) defined on the lattice. Furthermore, in the single sublattice model, due to the translation symmetry, $G_{M\sigma,ii} = 1 - n_\sigma$ is identical for all sites, we do further simplification on Eq. (37),

$$
\begin{aligned}
-\log|X_M^\rho| &= -\sum_\sigma \log|1 - \frac{\theta^2}{2} \sum_i (1 - G_{M\sigma,ii}) - \frac{\theta^2}{2} \sum_{\{i,j\}} (1 - G_{M\sigma,ii})(1 - G_{M\sigma,jj}) \\
&\quad - \frac{\theta^2}{2} C_{M,\sigma}^{2,latt} + i\theta \sum_i (1 - G_{M\sigma,ii}) + O(\theta^2)| \\
&= -\sum_\sigma \log\sqrt{(1 - \frac{\theta^2}{2} n_\sigma N_M - \frac{\theta^2}{2} n_\sigma^2 N_M (N_M - 1) - \frac{\theta^2}{2} C_{M,\sigma}^{2,latt})^2 + (\theta n_\sigma N_M)^2 + O(\theta^2)} \\
&= -\sum_\sigma \log\sqrt{1 - \theta^2 (n_\sigma N_M + n_\sigma^2 N_M (N_M - 1) + C_{M,\sigma}^{2,latt}) + \theta^2 n_\sigma^2 N_M^2 + O(\theta^2)} \\
&= -\sum_\sigma \log\sqrt{1 - \theta^2 (n_\sigma N_M - n_\sigma^2 N_M + C_{M,\sigma}^{2,latt}) + O(\theta^2)} \\
&= \frac{\theta^2}{2} \sum_\sigma \left( N_M (n_\sigma - n_\sigma^2) + C_{M,\sigma}^{2,latt} \right) + O(\theta^2)
\end{aligned}
\tag{38}
$$

Note $N_M$ in Eq. (38) is the number of site of region $M$, and $\hat{N}_M$ in Eq. (34) is the total particle number operator of region $M$. Without $C_{M,\sigma}^{2,latt}$, the disorder operator behaves as the volume law for any system. In addition, if $C_{M,\sigma}^{2,latt}$ is strictly zero at $r > r_0$, the behavior of is still volume law plus a constant at $r > r_0$. Generally speaking, various function form of disorder operator is determined by the function form of $C_{ij}$ and the shape of region $M$.

Compare Eq. (38) with Eq. (34), the two terms $N_M (n_\sigma - n_\sigma^2)$, $C_{M,\sigma}^{2,latt}$ in Eq. (38) correspond $r = 0$ and $r \neq 0$ part of density fluctuations $C^2$ in Eq. (34), respectively, since the summation for the latter requires $i \neq j$. Here we use $\hat{n}_i^2 = \hat{n}_i$ for the properties of fermionic particle density operator. As a consequence, we expect $C_{M,\sigma}^{2,latt} \approx C_{M,\sigma}^2$ at thermodynamic limit. Finally, we conclude the small $\theta$ expansion of the disorder operator describes the performance of two points correlation function.

#### 2. *large angle at $\theta = \pi/2$ and $2\pi/3$*

In non-interacting case, taking $\theta = \frac{\pi}{2}$ in $X_M^\rho$ and omit $\sigma$ index, we have,

$$
X_M^\rho(\frac{\pi}{2}) = \det\left[ G_M + i(\mathbb{1} - G_M) \right]
\tag{39}
$$

We display the exact relation of Eq.(8) and (9) in the main text,

$$
\begin{aligned}
S_2 &= -\log\{\det\left[G_M^2 + (\mathbb{1} - G_M)^2\right]\} \\
&= -\log\{\det\left[G_M + i(\mathbb{1} - G_M)\right] \times \det\left[G_M - i(\mathbb{1} - G_M)\right]\} \\
&= -\log\{X_M(\frac{\pi}{2})X_M(\frac{\pi}{2})^*\} \\
&= -\log|X_M(\frac{\pi}{2})|^2 \\
&= -2\log|X_M(\frac{\pi}{2})|.
\end{aligned}
\tag{40}
$$

And for $\theta = \frac{2\pi}{3}$,

$$
\begin{aligned}
S_3 &= -\frac{1}{2}\log\{\det\left[G_M^3 + (\mathbb{1} - G_M)^3\right]\} \\
&= -\frac{1}{2}\log\{\det\left[\mathbb{1} - 3G_M + G_M^2\right]\} \\
&= -\frac{1}{2}\log\{\det\left[\frac{1}{4}(\mathbb{1} - 3G_M)^2 + \frac{3}{4}(\mathbb{1} - G_M)^2\right]\} \\
&= -\frac{1}{2}\log\{\det\left[\frac{1}{2}(\mathbb{1} - 3G_M) + i\frac{\sqrt{3}}{2}(1 - G_M)\right] \times \left[\frac{1}{2}(\mathbb{1} - 3G_M) - i\frac{\sqrt{3}}{2}(1 - G_M)\right]\} \\
&= -\frac{1}{2}\log\{X_M(\frac{2\pi}{3})X_M(\frac{2\pi}{3})^*\} \\
&= -\frac{1}{2}\log|X_M(\frac{2\pi}{3})|^2 \\
&= -\log|X_M(\frac{2\pi}{3})|.
\end{aligned}
\tag{41}
$$

### 3. $\theta = \pi$

We emphasize that, in contrast with the bosonic system, the disorder operator defined in this form is differ from the entanglement entropy in the free system by taking $\theta = \pi$ in Ref. [26]. As shown above, one can strictly prove the equality between and the 2nd Rényi entropy between the disorder operator at $\frac{\pi}{2}$. For free fermion case, by taking $\theta = \pi$ in $X_M^\rho$, one obtain,

$$
-\log|X_M^\rho(\pi)| = -\log\det\left[-1 + 2G_M\right],
\tag{42}
$$

which gives divergence since the eigenvalue of $G_M$ has value of $0.5$.

### C. Section III: Density correlation function and disorder operator of several free fermionic system

We derive the density correlation function $C(\mathbf{r}) = C_{\mathbf{ij}}, r \neq 0$ at zero temperature analytically in several free 1D and 2D models with translation symmetry. Here $\mathbf{i,j}$ represent lattice site, $\mathbf{r} = \mathbf{i} - \mathbf{j}$, and $r = |\mathbf{r}|$. We set the length of unit cell in both 1D chain and 2D square lattice to be 1. We also calculate the density fluctuations $C^2$ in the region $M$ by the integral, which is the coefficient before small $\theta$ expansion. The shape of $M$ is chosen as Fig. 3(b) and (d) in the main text.

#### 1. Ground state of 1D fermions with FS

We use Hamiltonian Eq.(11) in the main text,

$$
\hat{H} = t_1 \sum_{\langle i,j \rangle} \hat{c}_i^\dagger \hat{c}_j + t_2 \sum_{\langle\langle i,j \rangle\rangle} \hat{c}_i^\dagger \hat{c}_j + \mu \sum_i \hat{n}_i
\tag{43}
$$

The density correlation function only depend on $k_F$,

$$
C(r) = \frac{g(-1 + \cos(2k_F r))}{2\pi^2 r^2}
\tag{44}
$$

where $g$ is fermion species. We find $C(r) \approx \frac{1}{r^2}$, and the global coefficient $(-1 + \cos(2k_F r))$ is related to $k_F$, which we identified as the oscillation term. We derivate the density fluctuation as following,

$$C^2 = \frac{g}{\pi^2} \log L_M + o(1). \tag{45}$$

Comparing Eqs. (44) and (45), the oscillation term do not effect the coefficient of leading term. And $\log L$ comes from $\frac{1}{r^2}$ relation for 1D.

### 2. *Ground state of 1D gapped fermionic system*

We use Hamiltonian as Eq. (43) added a staggered chemical potential, and written,

$$\hat{H} = t_1 \sum_{\langle i,j \rangle} \hat{c}_i^\dagger \hat{c}_j + t_2 \sum_{\langle\langle i,j \rangle\rangle} \hat{c}_i^\dagger \hat{c}_j + \Delta \sum_i (-1)^i \hat{c}_i^\dagger \hat{c}_i \tag{46}$$

Generally, the system consists of two sublattice. The density correlation at large distance writes,

$$C(r) \sim e^{-f(\Delta)r}, r \gg 1. \tag{47}$$

$f(\Delta)$ is a function of gap $\Delta$. The gapped physics drives the density fluctuation to converge to a constant at large scale,

$$C^2 = o(1), L_M \gg 1. \tag{48}$$

### 3. *Finite temperature of 1D fermionic system with fermi surface*

We still use Hamiltonian as Eq. (43), and study the density correlation at finite temperature, one have

$$C(r) \sim e^{-\xi r}(-1 + 2\cos(k_F r)), \tag{49}$$

where $\xi$ denotes the thermodynamics correlation length. Since exponential decay is convergent by the integral, the disorder operator is given as the volume law,

$$C^2 = L_M + o(1). \tag{50}$$

### 4. *Ground state of 2D fermions with FS*

Next, we study a 2D free fermion system, generated by the Hamiltonian Eq. (13) in the main text.

$$\hat{H} = t_1 \sum_{\langle i,j \rangle} \hat{c}_i^\dagger \hat{c}_j + t_2 \sum_{\langle\langle i,j \rangle\rangle} \hat{c}_i^\dagger \hat{c}_j + \mu \sum_i \hat{n}_i. \tag{51}$$

We discuss one simplest case, that is the circular FS, denoted by $k_F$. The density correlation function is only dependent on $r$. We have,

$$\begin{aligned} C(r) &= \frac{g k_F^2}{4\pi^2} \frac{J_1(k_F r)}{r^2} \\ &\approx \frac{g k_F}{4\pi^3 r^3}(1 - \sin(2k_F r)) \end{aligned} \tag{52}$$

$J_1$ represent the Bessel function. At large $r$, $C(r)$ obeys $\frac{1}{r^3}$ behavior with the oscillation coefficient. The leading term of the disorder operator has the well-known form $L_M^{d-1} \log L_M$, where $d$ is the dimension,

$$C^2 = \frac{2k_F}{\pi^3} L_M \log L_M + o(L_M) \tag{53}$$

Unlike the 1D FS, the coefficient of leading term is related to $k_F$, which can be easily understand use the conjectures in Ref [45]. The result is one special case equation of Eq. (9) in the main text. Since we fixed the region $M$ (FS) to be the square (circle), the double integral term is proportion to $k_F$. For general shape of fermi surface, we conclude the disorder operator written in Eq. (9) in the main text.

## D. Section IV: Fitting details and quantitative estimation in 2D free systems

In the main text, we discussed 1D and 2D free fermions with FS, where in 1D, we use the conformal distance $\tilde{L}_M$ to fit the Luttinger parameter. As the system size gets larger, the maximum of $\tilde{L}_M$ increase, and the Luttinger parameter fitted by the large $\tilde{L}_M$ gradually converge to the value of thermodynamic limit. Technically, one can push the system to large enough to get the accurate value. However, in 2D system, a good finite size modification parameter like $\tilde{L}_M$ is hard to find, and there is not one simple method to judge the convergence to thermodynamic limit of fitting results. As we mentioned in Fig.2 in the main text, we fit the function forms by certain fit range. Obviously, the choose of the fit range will change $s$ a lot. Therefore, we first explore the effect by the fit range in 2D, and compare with the analytic value, obtained by Eq. (53). We define the lower(upper) boundary of fit range as $f_{low}(f_{up})$, and fit the raw data with the function form $-\log|X_M(\theta)| = s_{2D}(\theta)L_M \log L_M + bL_M + c$ in (a) and $-\log|X_M(\theta)|/L_M = s_{2D}(\theta)\log L_M + b$, using $L_M \in [f_{low}, f_{up}]$ in Fig. 8. And we use at least 5 data points to fit, i.e. $f_{up} \geq f_{low} + 5$. From this two panels in Fig. 8, one can see the different original data brings distinguishing $s_{2D}$, where we find (b) is more possible to get the accurate value. We conclude the general fit principle, that one would choose $f_{low} \gg 1$, and $f_{up} \ll L$ to avoid finite size effect. That requires large system size up to at least hundreds of $L$. Besides, we observe that (b) has wider fit range to get same close value of analytic results. Thus, one may use $-\log|X_M|/L_M$ versus $\log L_M$ to decrease finite size effect.

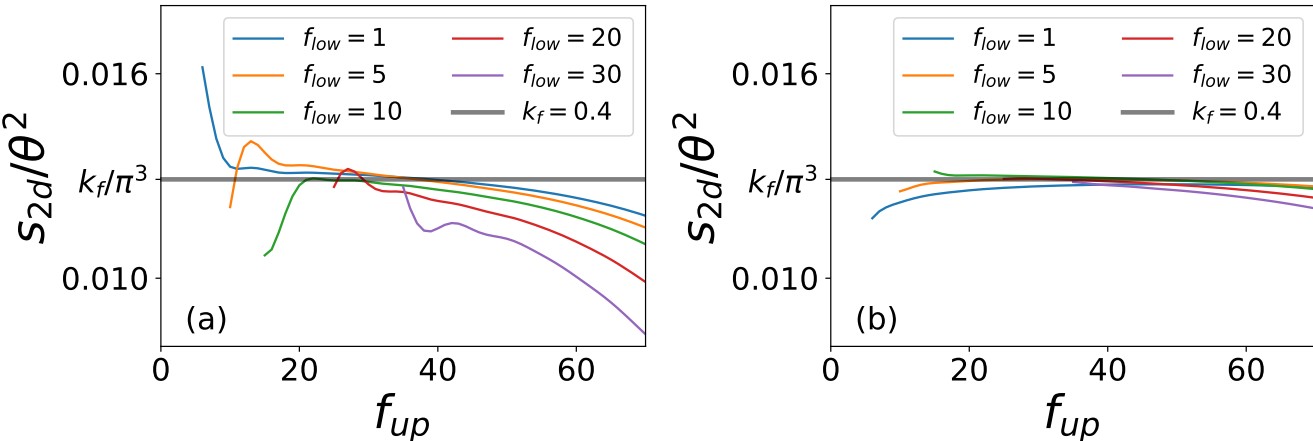

FIG. 8. Fit results with various fit range for free circular-shaped fermion at $L = 160$, $k_F = 0.4$. We choose few $f_{low}$ shown by different color. The analytic value is given by Eq. (53) the black solid line. The original data is (a) $-\log|X_M|$ fitted by $s_{2D}(\theta)L_M \log L_M + bL_M + c$, adapted in Fig. 2(a) in the main text. (b) $-\log|X_M|/L_M$ versus $\log L_M$ fitted by the linear function $y = s_{2D}x + b$, adapted in Fig. 2(b) in the main text. For comparison, (b) has more similar values for same fit range compared with analytic value $k_F/\pi^3$ given by Eq. (53).

## E. Section V: Analytical analysis of the disorder operator with Bosonization in 1D

We follow the convention that the left/right-moving fermion fields are bosonized as $\psi_{R/L}(x) \sim e^{-i[\pm\phi(x)-\vartheta(x)]}$. Using the expression for the charge density $\rho(x)$

$$\rho(x) = -\frac{1}{\pi}\partial_x\phi, \tag{54}$$

we can easily compute the disorder operator:

$$\left\langle \exp\left(i\theta \int_0^{L_M} dx\,\rho(x)\right)\right\rangle = \left\langle e^{\frac{i\theta}{\pi}[\phi(0)-\phi(L_M)]}\right\rangle \sim L_M^{-\frac{\theta^2 K}{2\pi^2}}. \tag{55}$$

Therefore

$$-\log|X_M(\theta)| \sim \frac{\theta^2 K}{2\pi^2}\log L_M + \cdots. \tag{56}$$

The non-interacting Hamiltonian corresponds to $K = 1$ and we indeed reproduce the result in Eq. (12).

Now let us generalize to interacting spin-$\frac{1}{2}$ electrons [63]. Following standard practice in bosonization we introduce

$$\phi_\rho(x) = \frac{1}{\sqrt{2}}[\phi_\uparrow(x) + \phi_\downarrow(x)],$$
$$\phi_\sigma(x) = \frac{1}{\sqrt{2}}[\phi_\uparrow(x) - \phi_\downarrow(x)],$$

(57)

and similarly $\vartheta_\rho$ and $\vartheta_\sigma$. The Hamiltonian now reads

$$H = \sum_{\alpha=\rho,\sigma} \frac{v_\alpha}{2\pi} \int dx \left[ K_\alpha(\partial_x \vartheta_\alpha)^2 + K_\alpha^{-1}(\partial_x \phi_\alpha)^2 \right].$$

(58)

Here $K_\rho$ and $K_\sigma$ are the Luttinger parameters of the charge and spin channels, respectively. Finally, we get,

$$
\begin{aligned}
-\log|X_M^\rho(\theta)| &= -\log|\langle e^{\frac{\sqrt{2}i\theta}{\pi}(\phi_\rho(0)-\phi_\rho(L_M))]}\rangle| \\
&= \frac{\theta^2 K_\rho}{\pi^2}\log L_M + \cdots,
\end{aligned}
$$

(59)

and similarly for the spin channel:

$$
\begin{aligned}
-\log|X_M^\sigma(\theta)| &= -\log|\langle e^{\frac{i\theta}{\sqrt{2}\pi}(\phi_\sigma(0)-\phi_\sigma(L_M))]}\rangle| \\
&= \frac{\theta^2 K_\sigma}{4\pi^2}\log L_M + \cdots,
\end{aligned}
$$

(60)

### F.   Section VI: Traditional way to determine the Luttinger parameter in 1D

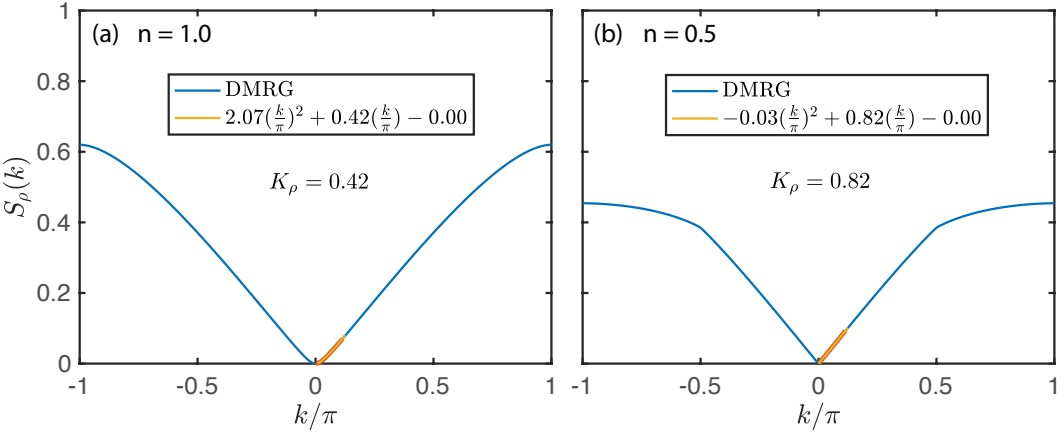

FIG. 9. In a $U=2$, $L=256$ Hubbard chain with open boundary condition, the charge structure factor $S_\rho$ is shown versus $k/\pi$, for both (a) half-filling case $n=1.0$ and (b) quarter-filling case $n=0.5$. The blue solid lines are the DMRG calculated data with the red dot depicting the data used for the extrapolation. The yellow solid lines indicates the 2nd order extrapolation $S_\rho = A(\frac{k}{\pi})^2 + B(\frac{k}{\pi}) + C$, from which the Luttinger charge exponent can be extracted $K_\rho = B = 0.42$ for the half-filling case, and $K_\rho = 0.82$ for the quarter-filling case.

In this section, we briefly recapitulate basic results from the Tomonaga-Luttinger liquid theory [112–114], on the charge correlations which constitutes an prevailing way to extract the Luttinger charge exponent $K_\rho$ numerically [66, 115, 116]. For the results listed below, we focus on the case with spin SU(2) symmetry and have the Luttinger parameter $K_\sigma = 1$ for the spin gapless states. In genenal, there exist multiple modes for the considered charge density correlation. Up to the first two dominant modes, we have

$$C(r) = -\frac{K_\rho}{(\pi r)^2} + A\frac{\cos(2k_F r)}{r^{1+K_\rho}}\ln^{-3/2}(r),$$

(61)

where A is a model-dependent parameters. The uniform mode (related to the $r^{-2}$ term above) results in $S_\rho(k) \simeq K_\rho|k|/\pi$ for $k \to 0$, with $S_\rho(k) = 1/L\sum_{i,j} e^{-ikr_{ij}}C_{ij}$ the Fourier transformation of the charge correlation.

In practice, for the charge gapless phase like LL, the asymptotic behavior of $S_\rho(k)$ is *linear* with $k$ as $k \to 0$, while in the charge-gapped phase we have $K_\rho = 0$ and the small-$k$ *quadratic* scaling. This offers us a way of extracting $K_\rho$ from charge density correlation. To be more specific, we employ a 2nd-order polynomial fitting $S_\rho(k) = A(k/\pi)^2 + B(k/\pi) + C$, from which the Luttinger charge exponent $K_\rho = B$.

The $S_\rho(k)$ results for $U = 2$ at the half-filling case of an $L = 256$ Hubbard chain with open boundary condition is shown in Fig. 9(a). Due to the exponentially small gap in the state, i.e., $\Delta_\rho \sim \exp^{(-t/U)}$, in the small-$k$ regime $S_\rho$ show a slightly quadratic behavior, and from the small-$k$ data (to eliminate the finite-size effect, here $k \in [2\pi/L, 2\pi n/L]$ with $n = 15$ chosen) we still get a finite value of $K_\rho = 0.42$ still far from the expected $K_\rho = 0$. For the quarter-filling case as shown in Fig. 9(b), the small-$k$ data show clear linear behavior and we have $K_\rho = 0.82$ in great agreement with the value in the main text and also Bethe ansatz result [65]. As we have shown in the main text Fig.4, we find it is much easier to fit the charge disorder operator and extract its coefficient – the Luttinger parameter – from the $K_\rho \log L$ scaling with much less finite size effects.