# Peer review of "Many versus one: the disorder operator and entanglement entropy in fermionic quantum matter"

_SciPost Physics_

## Round 4 · Referee Report · Anonymous (Referee 1) · 2023-5-12

Strengths

  1. It highlights the possibility of evaluating efficiently the disorder operator via quantum Monte Carlo in interacting 2d Fermi system, showing how it can be a good probe of the underlying phase of matter.
  2. Many known results of literature are nicely reviewed in a compact way

Weaknesses

  1. It is not clear what is the original content, and what is already known.
  2. The discussion about the two phases (paramagnetic and ferromagnetic) in the 2d system (5.C) is probably too compact with respect to the rest. It is a pity, as this subsection looks promising regarding possible new applications of the disorder operator.

Report

The general acceptance criteria are met, and I recommend it for publication. However, at first sight, the originality looks low, as most of the formula that are reported in the main text, especially the ones about 1 dimensional critical chains, are well-established (bosonization, Widom-Sobolev, R\'enyi entropy vs disorder operators,... ) and have little to do with the main message of this work. It would have been preferable to put less material, analyzing better the 2D Fermi/non-Fermi liquid case.

For this reason, I would ask the authors, to point out in a couple of sentences the new results separating them explicitly from the rest of the manuscript.

Requested changes

Below, I list some minor issues

  • After equation (5), what does it mean "the Green's function breaks down into spin up and down..."? It means that it (Eq. (4)) vanishes when \sigma \neq \sigma'? Probably one can write it down explicitly, to make it clearer.

  • After Eq. (7): "For non interacting-systems, det( ... ... ) can be factorized as det(...)det(...)..". This is just the Binet theorem about the determinant of the product, and it has nothing to do with non interacting-systems. Since it is a bit misleading, I would suggest to slightly rewrite the sentence, without referring explicitly to " non interacting-systems" at the beginning.

  • After Eq. (31). "For EE we expect S = c/4 log L + ... ". I think this is a typo and the replacement "c/4 -> c/3" is needed.

  • Page 2: "Subsequent studies focused on various application on certain systems, including fermioic system...". Fermioic -> Fermionic

  • validity: good
  • significance: good
  • originality: low
  • clarity: high
  • formatting: excellent
  • grammar: good

Author:  Bin-Bin Chen  on 2023-06-06  [id 3712]

(in reply to Report 1 on 2023-05-12)
Category:
remark

We appreciate the valuable comments and recommendation for publication by the respected reviewer.

Response to Report:
Reply: We thank the reviewer’s suggestion of making the 2D results more prominent. Indeed, our intention was to show the potential of the disorder operator as an efficient and valuable tool for probing the phases of matter in interacting 2D fermion systems. The positive feedback from the reviewer assures us that we’ve been successful in conveying this message. We fully agree with the reviewer that we shall consolidate and review existing knowledge in a more concise manner, and have revised the manuscript to make our original contributions more prominent and to place a greater focus on the 2D Fermi/non-Fermi liquid cases. We’ve also explicitly highlighted our new results in a separate section in Sec.II “Summary of key results” of the revised manuscript, and we’ve reduced the exposition of well-known concepts.

Response to weakness 1 : "It is not clear what is the original content, and what is already known."
Reply: We apologize for any lack of clarity in differentiating between our original contributions and existing knowledge. In response to the comment, we have revised the manuscript to clearly delineate our original findings from the prior works. We’ve added references to previous works wherever necessary to indicate established knowledge and have added a section in Sec.II “Summary of key results” of the revised manuscript to explicitly highlight our original findings.

Response to weakness 2: "The discussion about the two phases (paramagnetic and ferromagnetic) in the 2d system (5.C) is probably too compact with respect to the rest. It is a pity, as this subsection looks promising regarding possible new applications of the disorder operator."
Reply: We thank the respected reviewer for the interests on the discussion of the two phases in the 2D system. Originally, since the phase diagram from paramagnetic through non-Fermi-liquid to ferromagnetic phases is from our previous works, we therefore have shortened their description in our previous manuscript. In response to the valuable suggestion, in the revised manuscript, we have expanded the discussion in subsection 5.C, providing a more detailed explanation of the paramagnetic and ferromagnetic phases and their implications. We believe that this revision provides a more balanced and informative presentation of our findings in the 2D Fermi liquid and non-Fermi-liquid quantum critical metal systems.

Response to Requested change 1: After equation (5), what does it mean "the Green's function breaks down into spin up and down..."? It means that it (Eq. (4)) vanishes when $\sigma \neq \sigma'$? Probably one can write it down explicitly, to make it clearer.
Reply: We thank the reviewer for the careful reading of our paper. We intended to express that the Green's function matrix is block-diagonal and can be separated into two parts, {and we change the sentence to ``the Green's function matrix is block-diagonal with two sectors, i.e., the spin up sector $G_{M,\uparrow}$ and the spin down sector $G_{M,\downarrow}$.''

Response to Requested change 2 : After Eq. (7): "For non interacting-systems, det( ... ... ) can be factorized as det(...)det(...)..". This is just the Binet theorem about the determinant of the product, and it has nothing to do with non interacting-systems. Since it is a bit misleading, I would suggest to slightly rewrite the sentence, without referring explicitly to " non interacting-systems" at the beginning.
Reply : We thank the respected reviewer for the careful reading of our paper. We follow the suggestion and rewrite the sentence to avoid possible confusion. But we would also like to explain the situation a bit more clearer here. As for the Eq.(7), there is indeed a difference between the non-interacting system and interacting one. For the latter, the auxiliary field {$s$} is introduced, the expression of the charge channel disorder operator is written as Eq. (10),
\begin{equation}
X_M^{\rho}(\theta) = \sum_{\{ s \}} P_s \text{det} \left[G_{M,s} + (I - G_{M,s}) e^{i\theta} \right],
\end{equation}
In the interacting systems we considered, the block-diagonal structure of $G_{M,s}$ still holds. Thus one can only write
\begin{equation}
X_M^{\rho}(\theta) = \sum_{\{ s \}} P_s \text{det} \left[ \left( G_{M,s,\uparrow} + (I - G_{M,s\uparrow}) e^{i\theta} \right) \times \left( G_{M,s,\downarrow} + (I - G_{M,s,\downarrow}) e^{i\theta} \right) \right]
\end{equation}
but not in this form,
\begin{equation}
\begin{aligned}
X_M^{\rho}(\theta) &\neq X_{M,\uparrow}(\theta) X_{M,\downarrow}(\theta) \\
&\equiv \sum_{\{ s \}} P_s \text{det} \left( G_{M,s,\uparrow} + (I - G_{M,s\uparrow}) e^{i\theta}\right) \times \sum_{\{ s \}} P_s \text{det} \left( G_{M,s,\downarrow} + (I - G_{M,s,\downarrow}) e^{i\theta} \right)
\end{aligned}
\end{equation}
Note the definitions of $X_{M,\uparrow}, X_{M,\downarrow}$ are also extended to the interacting case. Only in the non-interacting case without auxiliary filed, the inequality above becomes equality sign.

Response to Requested change 3: After Eq. (31). "For EE we expect S = c/4 log L + ... ". I think this is a typo and the replacement "c/4 -$>$ c/3" is needed.
Reply : Here the entanglement entropy we considered is second-order Renyi entropy, and the coefficient of the logarithmic term is $c/4$ other than $c/3$. To avoid confusion, we change the sentence into ``For EE (Renyi entropy $S_2$), we expect $S_2=\frac{c}{4} \log{L_M} + \text{const.}$''.

Response to Requested change 4: Page 2: "Subsequent studies focused on various application on certain systems, including fermioic system...". Fermioic -$>$ Fermionic
Reply: We thank the reviewer for the careful reading and have fixed this typo in the revised manuscript.

---

## Round 4 · Referee Report · Anonymous (Referee 2) · 2023-5-27

Strengths

The paper discusses connections between entanglement of fermionic many body states with conserved charge and the disorder operator. In free fermion systems new identities are proven. In interacting systems in two dimensions in which much less is known a connection to quantum monte Carlo data is shown allowing to study exotic 2D critical points from the point of view of entanglement.

Weaknesses

The paper could be better organized and shortened, in a way that allows to immediately understand what are the novel results derived in this paper as compared to what is known.

In my view the 1D examples occupy too much space pushing the more novel 2D interacting cases to the end pf the paper.

Report

The relation between quantum entanglement and the disorder operator, which is the exponent of the subsystem number operator, are explored. The various results including exact relations to the Renyi entropy in general free fermion systems and the relation to QMC in the presence of interactions and even beyond one dimension, are novel and interesting. Thus I think the paper is an important contribution to the field of many-body entangled phases, and I recommend it to be published in Sci|Post, provided that the remarks below are addressed.

Requested changes

The grammar should be fixed everywhere! Example (already in the 1st line in the abstract)
... the disorder operator successfully probe... --> ... the disorder operator successfully probes...

Optional comment: While the paper deals with entanglement entropies which characterize zero temperature pure states, it is interesting whether the relationship between the disorder operator and entanglement extends to measures of mixed states such as negativity or specifically number entanglement which is based on the conserved charge, see for example PRL 130, 136201 (2023).

  • validity: top
  • significance: high
  • originality: high
  • clarity: high
  • formatting: perfect
  • grammar: acceptable

Author:  Bin-Bin Chen  on 2023-06-06  [id 3710]

(in reply to Report 2 on 2023-05-27)
Category:
remark

We thank the respected reviewer for the supportive comments and valuable feedback on our manuscript.

Response to Report:
We are very glad that the respected reviewer find the discussions and results in our paper valuable. Indeed, we share the same point of view that the connections between entanglement and the disorder operator in both free and interacting fermionic systems can provide a novel perspective for the study of quantum many-body entanglement. Your suggestions regarding the presentation and structure of the paper have greatly helped us in improving our manuscript.

Response to Weakness:
We thank the reviewer for the valuable suggestion and understand the need for making our new results more prominent. To address this issue, in the revised introduction, we now clearly state the distinction between our new results and existing knowledge. Furthermore, we have also made efforts to shorten the paper by eliminating redundancies and streamlining the presentation of our findings. Our initial intention was to build a solid understanding using the 1D cases before revealing the more complex 2D results. However, we follow the suggestion of the reviewer, and shortened the 1D examples section and move some of the derivations into the Supplementary Materials. We have also moved up the 2D interacting cases to ensure that they gain appropriate attention. This also adds to the conciseness of the paper without sacrificing important information.

Response to Requested changes:
We sincerely apologise for the grammatical errors present in our manuscript. We have carefully revised the entire manuscript for grammatical mistakes and typos, including the specific example from the abstract, which has been corrected to “...the disorder operator successfully probes...”. We believe these changes have improved the readability of our paper.

Response to Optional Comment:
We thank the respected reviewer for the constructive suggestions and for pointing us to this important PRL work, which we have explicitly cited in the revised manuscript.
The extension of the relationship between the disorder operator and entanglement to mixed states is indeed an interesting question. Although our current study does not directly address this extension, we appreciate the valuable suggestion and accordingly include a discussion on the extension to mixed states in the revised manuscript. In the light of that, we will certainly consider these aspects in our future works.

---

## Editorial Decision

resubmitted